# Behavioral entrainment to rhythmic auditory stimulation can be modulated by tACS depending on the electrical stimulation field properties

Yuranny Cabral-Calderin[1]*, Daniela van Hinsberg[1], Axel Thielscher[2,3], Molly J Henry[1,4]*

[1]Max Planck Institute for Empirical Aesthetics, Frankfurt, Germany; [2]Danish Research Centre for Magnetic Resonance, Centre for Functional and Diagnostic Imaging and Research, Copenhagen University Hospital Amager and Hvidovre, Copenhagen, Denmark; [3]Section for Magnetic Resonance, DTU Health Tech, Technical University of Denmark, Copenhagen, Denmark; [4]Toronto Metropolitan University, Toronto, Canada

*For correspondence:
yuranny.cabral-calderin@ae.mpg.de (YC-C);
molly.henry@ae.mpg.de (MJH)

Competing interest: The authors declare that no competing interests exist.

**Abstract** Synchronization between auditory stimuli and brain rhythms is beneficial for perception. In principle, auditory perception could be improved by facilitating neural entrainment to sounds via brain stimulation. However, high inter-individual variability of brain stimulation effects questions the usefulness of this approach. Here we aimed to modulate auditory perception by modulating neural entrainment to frequency modulated (FM) sounds using transcranial alternating current stimulation (tACS). In addition, we evaluated the advantage of using tACS montages spatially optimized for each individual's anatomy and functional data compared to a standard montage applied to all participants. Across two different sessions, 2 Hz tACS was applied targeting auditory brain regions. Concurrent with tACS, participants listened to FM stimuli with modulation rate matching the tACS frequency but with different phase lags relative to the tACS, and detected silent gaps embedded in the FM sound. We observed that tACS modulated the strength of behavioral entrainment to the FM sound in a phase-lag specific manner. Both the optimal tACS lag and the magnitude of the tACS effect were variable across participants and sessions. Inter-individual variability of tACS effects was best explained by the strength of the inward electric field, depending on the field focality and proximity to the target brain region. Although additional evidence is necessary, our results also provided suggestive insights that spatially optimizing the electrode montage could be a promising tool to reduce inter-individual variability of tACS effects. This work demonstrates that tACS effectively modulates entrainment to sounds depending on the optimality of the electric field. However, the lack of reliability on optimal tACS lags calls for caution when planning tACS experiments based on separate sessions.

## eLife assessment

This detailed and well powered manuscript explores auditory perception of modulated noise in the presence of transcranial alternating-current stimulation (tACS) and shows **valuable** results suggesting that there are subject-specific effects when the phase of 2-Hz tACS varies relative to the phase of the noise modulation. The strength of the evidence is mixed. There is **convincing** evidence that tACS alters perception significantly in individuals; however, the effects are inconsistent across subjects and even across sessions, frustrating attempts to draw conclusions about the underlying mechanisms of the idiosyncratic effects. Despite these limitations, the paper will be of great interest

to researchers interested in determining when and how tACS influences neural processes, especially those interested in neural entrainment and its relationship to perception.

## Introduction

Brain rhythms synchronize with – entrain to – rhythms in sounds, and this stimulus-brain synchrony potentially facilitates perception and comprehension of auditory sensory information (*Peelle and Davis, 2012*; *Zion Golumbic et al., 2013*; *Horton et al., 2013*; *Doelling et al., 2014*; *Horton et al., 2014*; *Doelling and Poeppel, 2015*; *Nozaradan et al., 2016*; *Brodbeck et al., 2020*). For example, successful neural entrainment seems to support cocktail-party listening – in a cocktail party scenario, speech tracking in and near low-level auditory cortices is enhanced for attended vs. ignored speech, and only attended speech has been observed to be tracked in higher order brain regions (*Zion Golumbic et al., 2013*). In addition, stronger neural entrainment to the beat in music has been associated with superior beat prediction abilities and with better movement synchronization accuracy with a rhythmic beat (*Nozaradan et al., 2016*). Thus, stronger neural entrainment to auditory rhythms seems to support successful and adaptive listening.

If entrainment to auditory rhythms is really beneficial for auditory perception, then improving entrainment via external manipulations should lead to corresponding improvements in auditory perception. One technique often used to interfere with neural entrainment is transcranial alternating current stimulation (tACS). During tACS, a low-intensity alternating current is applied to scalp electrodes with the aim of reaching the brain and entraining brain activity to the electrical stimulation (*Herrmann et al., 2013*; *Reato et al., 2013*; *Cabral-Calderin and Wilke, 2020*). The efficacy of tACS-entrainment has been supported by studies showing increased power in EEG signals at the frequency of the electrical stimulation as well as phase alignment of intrinsic brain activity to the applied alternating current in single-unit recordings (*Krause et al., 2019*; *Johnson et al., 2020*).

In the auditory domain, tACS has been shown to significantly modulate speech comprehension, stream segregation, and binaural integration (*Riecke et al., 2015*; *Riecke et al., 2018*; *Wilsch et al., 2018*; *Zoefel et al., 2018*; *van Bree et al., 2021*; *Zoefel et al., 2020*; *Preisig et al., 2021*). For example, 4 Hz tACS modulates segregation of an auditory stream from background in a cyclic manner (*Riecke et al., 2015*). In addition, tACS synchronized with the speech envelope of attended as well as ignored speech signals in a cocktail-party scenario modulates speech comprehension of the target speech, albeit with opposite phase relationship (*Keshavarzi et al., 2021*).

Despite the promising findings, inconsistent results across studies have led to questions regarding the robustness, reliability, and reproducibility of tACS effects (*Erkens et al., 2020*; *Coldea et al., 2021*). Intra-individual reliability of tACS effects is often overlooked since multi-session tACS studies, where the same tACS protocol is repeatedly tested, are scarce. In addition, tACS effects have been shown to be state-dependent and variable across participants (*Kasten et al., 2019*; *Cabral-Calderin et al., 2016a*). Inter-individual variability on tACS effects might also arise from variability in brain anatomy, leading to inter-individual variability in the electric field induced by tACS when using the same electrodes montage and current strength (*Evans et al., 2020*). Indeed, inter-individual variability in electric field strength predicts inter- individual variability in tACS effects (*Kasten et al., 2019*; *Cabral-Calderin et al., 2016b*; *Zanto et al., 2021*). Thus, using the same tACS parameters (frequency, current strength, electrode position) might not be optimal for every participant or for a given participant under different conditions. Rather, optimal tACS protocols might need to be tailored to each participant to target the right frequency (*Stecher and Herrmann, 2018*) or stimulus-brain lag (*van Bree et al., 2021*) in the relevant brain region. Yet, studies testing the benefits of individually optimized tACS montages are scarce in the field.

In the present study, we test (1) the efficacy of tACS for modulating auditory perception, (2) the benefits of using individually optimized tACS montages for modulating behavior, and (3) the within-participant reliability of tACS results. Across two different sessions, participants listened to a frequency modulated (FM) noise stimulus and detected silent gaps presented at different phase positions within an FM cycle, while tACS was applied at different phase lags relative to the FM stimulus. Using structural and functional magnetic resonance imaging (fMRI), we optimized the tACS montage to target individually defined functional targets and tested its superiority over a group montage both in terms of the induced electric fields and the magnitude and variability of tACS effects on performance.

Specifically, we ask 4 questions: (1) can tACS at different phase lags interfere with stimulus-brain synchrony and modulate the behavioral signatures of entrainment? (2) are tACS effects reliable over time? (3) what factors predict individual variability of tACS effects? And (4) do tACS montages individually optimized in terms of spatial location yield stronger modulations of auditory perception?

## Results

### Using fMRI and FEM models to individually optimize tACS montages

Using one-size-fits-all montages for tACS can lead to high inter-individual variability in terms of the induced electric field and the strength of tACS effects. Here, we go a step past the current literature by individualizing the tACS montages to target individually defined functional targets and compare their electric field properties relative to two different group montages previously used in the literature for targeting auditory regions (*Figures 1–2*; *van Bree et al., 2021*; *Baltus et al., 2018*).

Listeners participated in one magnetic resonance imaging (MRI) session where functional and structural MRI datasets were collected. During the functional runs, participants listened to a 2 Hz FM stimulus and detected silent gaps presented at different phase locations along the modulation cycle (*Figure 1a*, see Materials and methods). As expected, increased BOLD signal in auditory temporal areas, including bilateral Heschl's gyri and superior temporal gyrus, was observed in response to the auditory stimulus (*Figure 1b*). Target regions of interest (ROI) were defined for each individual as the brain regions exhibiting the strongest response to the FM-stimulus (one per hemisphere, see Materials and methods, *Figure 1c and e*, *Supplementary file 1a*).

Using individual finite element method (FEM) head models (see Materials and methods) and the lead field-based constrained optimization approach implemented in SimNIBS (*Saturnino et al., 2019*), the individual electrode montage was optimized to focus the electric field as precisely as possible at the target ROI, with a field orientation matching an auditory dipole as estimated in a previous study targeting auditory regions (*Baltus et al., 2018*), and a desired field intensity of 0.09 V/m (*Figure 1d–g*, see Methods). The resulting individual montages varied across participants and could take on two out of seven different electrode positions for the left hemisphere: FC5, C1, C3, C5, T7, TP7, P7, and two out of nine different electrode positions for the right hemisphere: FC4, FC6, C2, C4, C6, T8, TP8, P8, and PO8. Despite some variability, the most common montages included the electrodes C5-TP7 and C6-TP8, with for example TP7 being used for 34 out of 39 participants (*Figure 1f*).

Electric field simulations were performed for each individual brain using the individually optimized montage (from now on referred to as *individualized montage*) with electrodes placed differently across participants, as well as two additional montages from the literature, with electrodes placed at the same positions across participants (*Figure 2a*, see Materials and methods): (1) same electrode shape and size as for the *individualized montage* but placed over FC5-TP7/P7 and FC6-TP8/P8 (referred to as *standard montage* in the rest of the document) as in *Baltus et al., 2018*, and a *ring-electrode montage* with center electrodes placed over T7 and T8 similar to the montage previously used by *van Bree et al., 2021*.

To compare between tACS montages in terms of their simulated electric field profiles, seven different dependent measures were computed for each participant and montage. Two measures referred to spatial targeting: (1) mean Euclidian distance between the center coordinates of the functional ROIs (one per hemisphere) and the center coordinates of the peak of the electric field in each hemisphere (where the peak is for the 99-percentile-thresholded E-field strength across all voxels for the given hemisphere) and (2) spatial Pearson correlation coefficient between the individual BOLD signal t-map and the electric field map. Four measures quantified the electric field (E-field) strength: (3) peak E-field strength for the whole brain (based on the 95-percentile-thresholded E-field map), (4) mean peak E-field strength in the functional ROIs (based on the 95-percentile-thresholded E-field from a 7 mm sphere around the center coordinates of the functional ROIs), (5) peak normal component of the E-field for the whole brain (95-percentile-thresholded), and (6) peak normal component of the E-field in the functional ROI (defined as in 4). The last measure quantified (7) E-field focality (cortical area in mm² with E-field strength higher than the 50th percentile). Repeated measures analysis of variance (rmANOVA) showed a significant effect of montage on all of the investigated measures (all $F_{(2,76)} \geq 9.938$, $p \leq 1.5\text{e-}04$, *Figure 2b*).

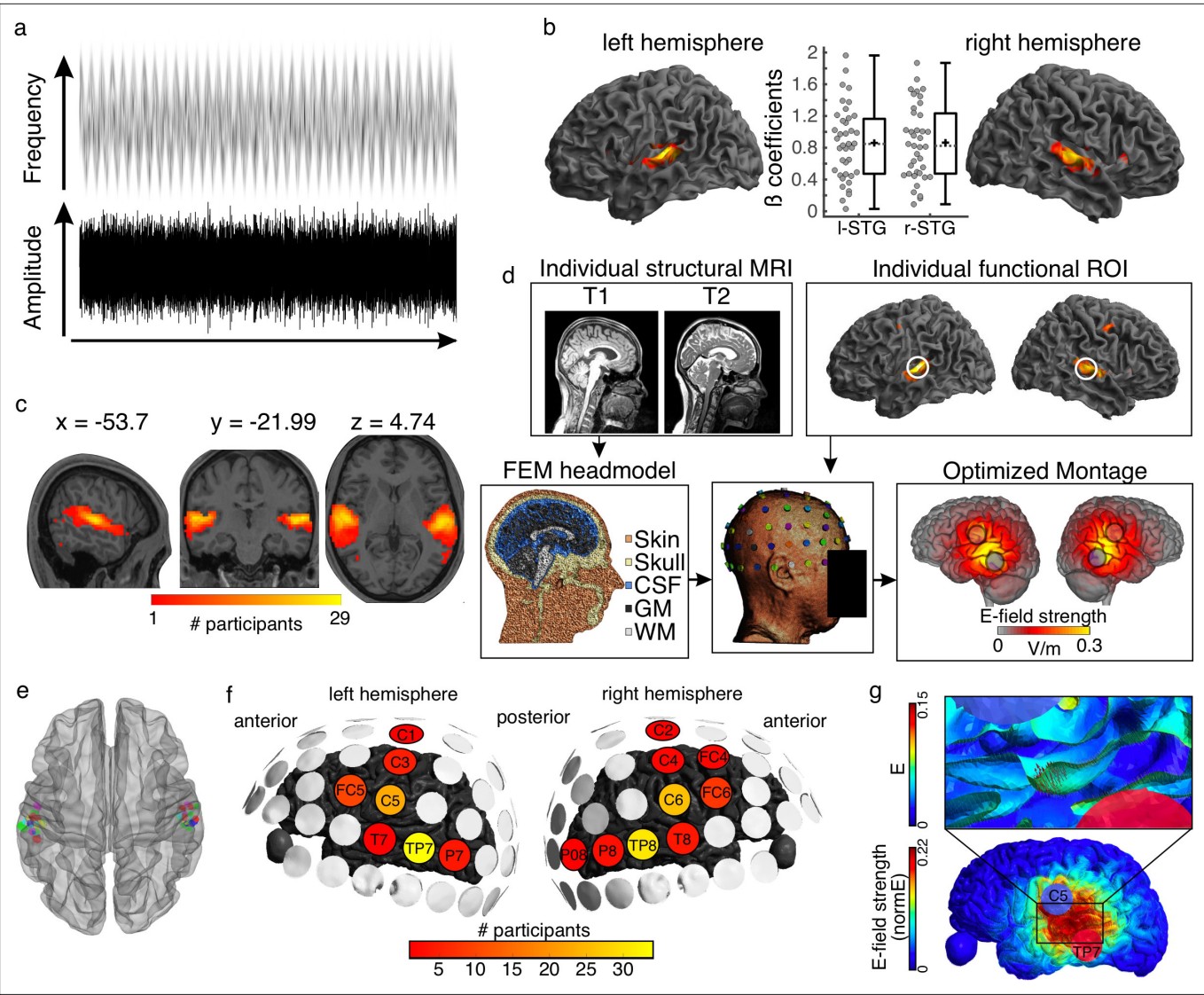

**Figure 1.** Auditory stimulus and tACS optimization pipeline. (**a**) Stimulus representation. A complex noise stimulus is frequency modulated at 2 Hz without any rhythmic modulation of its amplitude. Silent gaps are presented at different phase locations of the FM-stimulus modulation cycle. (**b**) Group data showing the regions exhibiting higher BOLD signal during the FM-stimulus presentation compared to the implicit baseline, p(FWE)<0.05. Graphs in the center show the beta estimates extracted for the whole cluster for each participant per hemisphere. Box plots show median (horizontal dashed lines), mean (black cross), 25th and 75th percentiles (box edges) and extreme datapoints not considered outliers (+/–2.7σ and 99.3 percentiles, whiskers). Each circle represents a single participant, N = 39. (**c**) Overlap of single-participant binary masks after thresholding the individual t-maps for the same contrast as shown in (**b**). (**d**) Pipeline for optimizing the tACS electrode montage for each individual participant to target the individual functional targets. (**e**) Target regions of interest used for the optimization step in SimNIBS. Individual dots represent the individual 3-mm-radius spheres around the center coordinates from the functional masks shown in (**c**). (**f**) Electrodes included in the optimized montages across participants. (**g**) Electric field (**e**) and electric field strength (normE) resulting from the optimized montage for one example participant. Only the left hemisphere is shown. Blue and red circles denote the resulting electrodes. Small red arrows on the inset show the target E-field orientation.

Regarding the spatial targeting of the functional ROI, the Euclidian distance between the E-field peaks and the target functional ROIs averaged across hemispheres was higher for the ring-electrode montage compared to both the individualized and the standard montages (all $t_{(38)} \geq 4.986$, $p \leq 4.163e-05$, *Bonferroni corrected*). Fisher-Z-transformed correlation values between the E-field map and the BOLD signal map were also higher for the ring-electrode montage compared to both individualized and standard montages (all $t_{(38)} \geq 2.798$, $p \leq 0.024$, *Bonferroni corrected*). Thus, the ring-electrode montage produced electric fields that were better correlated with the functional activation maps across the entire brain than the other two montages, but peak electric field missed the mark.

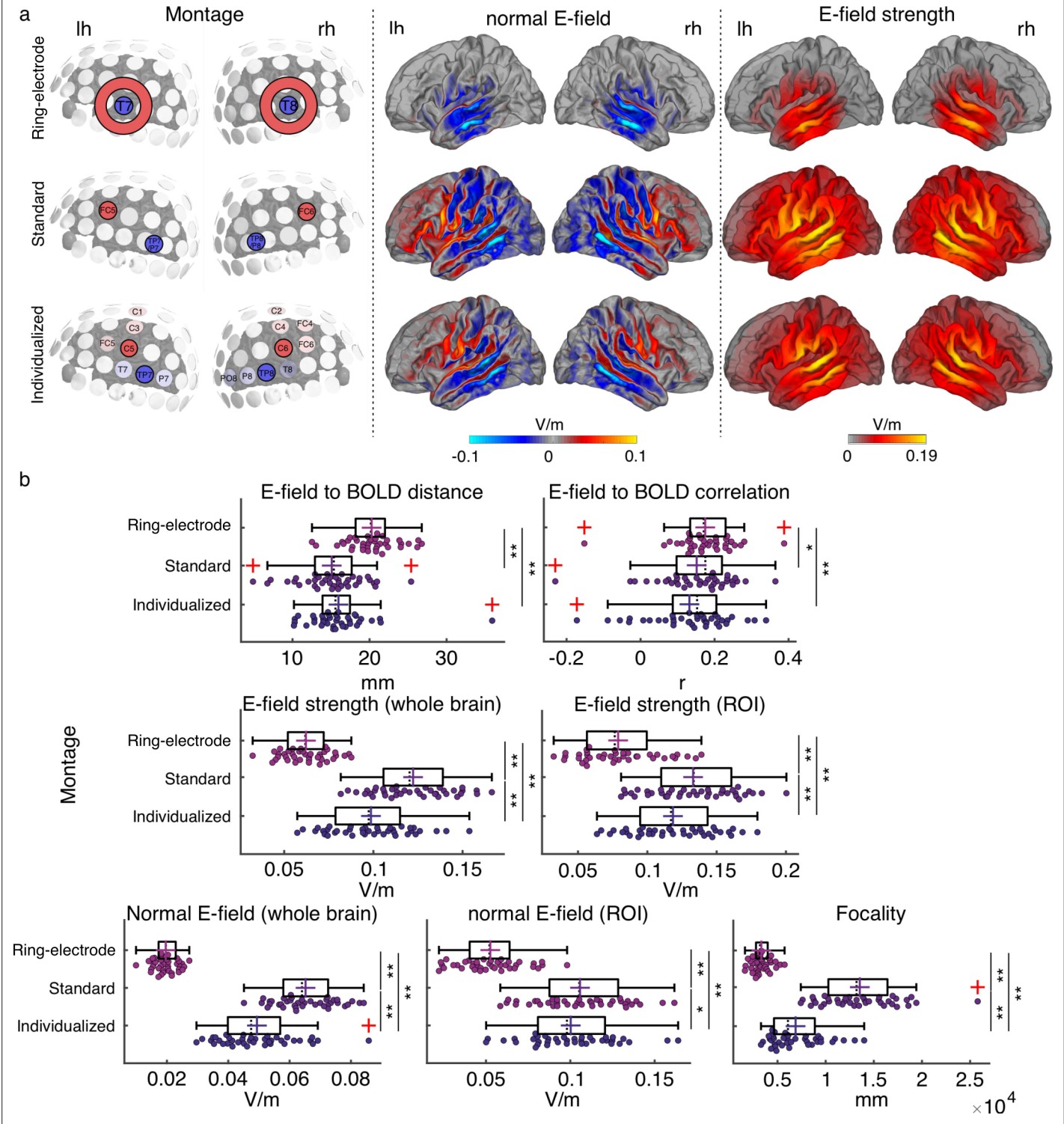

**Figure 2.** Electric field simulation results. (**a**) Group average maps showing the strength of the simulated electric field (E-field strength, right) and its normal component (normal E-field, middle), separated by montage: ring-electrode montage, top; standard montage, middle, and individualized montage, bottom. Each montage is represented in the left subpanel. (**b**) Plots showing the individual values for the seven E-field parameters estimated per montage and participant. Each dot represents a single participant. Box plots show median (dashed vertical lines), mean (cross in the middle of the box), 25th and 75th percentiles (box edges) and extreme datapoints not considered outliers (+/–2.7σ and 99.3 percentiles, whiskers). Red crosses represent outliers (more than 1.5 times the interquartile range away from the bottom or top of the box). Note that outliers were not excluded from analyses. *p < 0.05, **p < 0.001, post-hoc paired-samples t-tests, Bonferroni corrected.

No significant difference between the standard and the individualized montage was observed either for the distance between the E-field and the functional map peaks or the E-field to BOLD correlation (all $|t|_{(38)} \leq 2.047$, p $\geq 0.143$, *Bonferroni corrected*).

Regarding E-field strength, the standard montage induced higher electric field than the ring-electrode montage and the individualized montages for the whole brain and within the ROI; the same was true when considering only its normal component (all $t_{(38)} \geq 2.568$, p$\leq$0.043, *Bonferroni corrected, Figure 2b*). The individualized montage showed stronger E-field and normal component of the E-field, for the whole brain and within the ROI, when compared to the ring-electrode montage (all $t_{(38)} \geq 13.43$, p $\leq 1.665$e-15, *Bonferroni corrected, Figure 2b*). In terms of focality, the ring-electrode montage was the most focal of the three (all $t_{(38)} \geq 8,098$, p $\leq 2.54$e-9, *Bonferroni corrected, Figure 2b*), followed by the individualized montage which was more focal than the standard ($t_{(38)} = 9.578$, p = 3.349e-11, *Bonferroni corrected, Figure 2b*). Thus, each montage has its pros and cons, and the choice of montage will depend on which of these dependent measures is prioritized. The critical question then is which measure – or combination of measures – best predicts inter-individual variability in the magnitude of the tACS effect on behavior.

Before running any simulations, we had decided a priori to use only the individualized and standard montages for our main behavioral experiment (see below), motivated by the fact that the 4-electrode arrangement allows for directing the E-field in a desired orientation (*Baltus et al., 2018*). Targeting a specific field orientation would not be possible using the ring-electrode configuration from the ring-electrode montage. In addition, comparing between the simulated E-fields across montages showed that the ring-electrode montage with the specific location tested here might in fact not be optimal for modulating oscillatory activity during our auditory task, since the peak of the induced E-field was farther away from our target ROI and its strength was weaker compared to that simulated for the other two montages. As already described, we did not observe any difference between the individual and the group montages in terms of spatial overlap to our target ROI. While the electric field was strongest for the standard montage compared to the individualized montage, the latter induced a more focal field compared to the standard montage, therefore each montage could be good in different ways. In the behavioral experiments included in the rest of the Results section, only the individualized and the standard montage were used. The ring-electrode montage was not tested any further.

## FM-stimulus modulates gap detection with similar strength and optimal phase across participants and sessions

Across two different sessions, subjects listened to 2 Hz FM sounds and detected silent gaps presented at different phase locations on the FM modulation cycle. While performing the task, participants received either sham or verum tACS at 2 Hz (within participant) with either the individualized montage or the standard montage (between participants). During the verum tACS runs, the phase lag between the auditory stimulus and the tACS signal varied from trial to trial (*Figure 3a*). Participants missing one session (N = 5) were excluded at this stage. All further analysis is presented for 37 participants (individualized montage: N = 16, standard montage: N = 21).

To evaluate the effect of the FM-stimulus phase on gap detection in the absence of tACS, gaps presented during the sham condition were binned according to the FM-stimulus phase into which the gap fell and the mean detection rate (hit rate) was computed for each bin (*Figure 3b*). The strength of the FM-stimulus-driven behavioral modulation (*FM-amplitude*) and the optimal FM-stimulus phase (*FM-phase*) for gap detection were estimated by fitting a cosine function to hit rates as a function of FM-stimulus phase (*Cabral-Calderin and Henry, 2022*). *FM-amplitude* was significantly higher than chance for 33/37 participants in session 1 and for 27/37 participants in session 2 (comparison between each participant's *FM-amplitude* and its corresponding surrogate distribution, p < 0.05, *Figure 3c*). In addition, *FM-phase* was clustered across participants (Rayleigh test session 1: $Z = 29.031$, p = 1.806e-17; session 2: $Z = 26.531$, p = 2.12e-15). No significant difference between sessions was observed either for *FM-amplitude* ($t_{(36)} = 0.99$, p = 0.329) or for *FM-phase* (inter session phase distance clustered around zero, $V = 28.391$, p = 2.045e-11). A significant correlation between sessions was observed for *FM-amplitude* (Pearson's r = 0.516, p = 0.001). The latter replicates our previous finding that the FM-stimulus driven modulation of gap detection performance is reliable across time. We retained the *FM-amplitude* and *FM-phase* parameters as reliable signatures of behavioral entrainment to the FM-stimulus and asked whether and how these signatures are modulated by tACS.

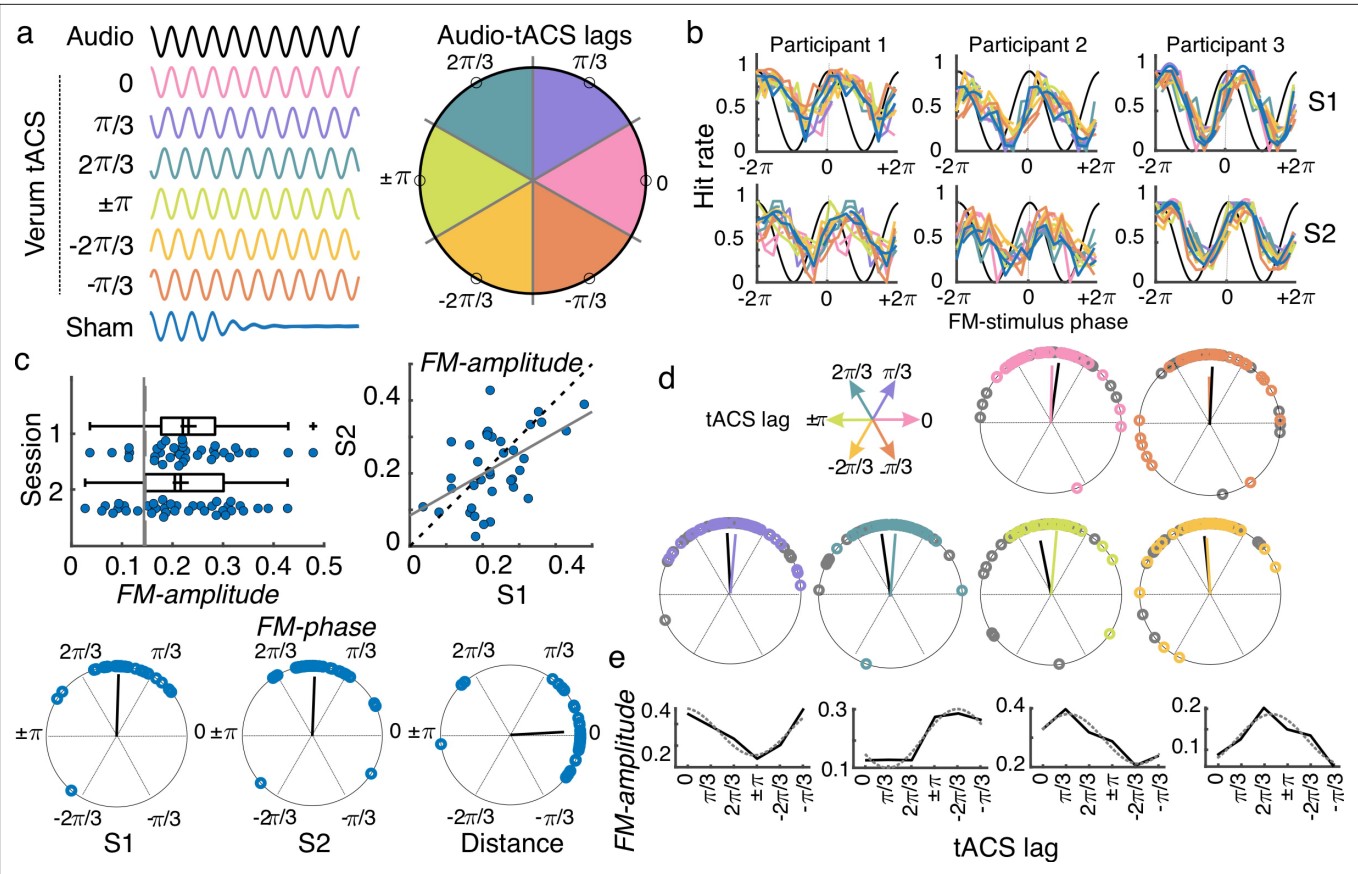

**Figure 3.** tACS effects and individual variability. (**a**) While performing the auditory task, participants received either sham (blue) or 2 Hz tACS stimulation. The phase lag between the FM stimulus (black) and the tACS signal varied from trial to trial and was grouped into six different phase-lag bins. Each color in the figure represents a different bin. Empty black circles in the circular plot on the left mark the phase in the center of each bin. (**b**) Hit rates as a function of FM-stimulus phase separated by tACS condition for both sessions (S1 and S2) from three example participants. Colors follow the same coding as in (**a**). (**c**) Amplitude of the FM-stimulus driven behavioral modulation (FM-amplitude) and optimal FM-stimulus phase (FM-phase) in the sham condition. Left top plot shows the FM-amplitude values from the sham condition. Vertical lines represent the mean 95 percentile from individual surrogate distribution, session 1 dashed line, session 2 solid line. Each dot represents a single participant. Box plots show median (dashed vertical lines), mean (cross in the middle of the box), 25th and 75th percentiles (box edges) and extreme datapoints not considered outliers (+/–2.7σ and 99.3 percentiles, whiskers). Crosses represent outliers (more than 1.5 times the interquartile range away from the bottom or top of the box). Scatter plot on the right top shows the FM-amplitude in session 2 (S2) as a function of FM-amplitude in session 1 (S1). Dashed line is the diagonal and the solid line the best-fit regression line. Circular plots in the bottom panel show the optimal FM-phase for each session and the circular distance between sessions. The black line is the resultant vector. (**d**) Optimal FM-phase separated by tACS lag condition. Each plot shows a different phase lag according to the coordinates presented in top left legend. Color code also matches that from panel (**a**). For each plot, session 1 is presented in the corresponding color and session 2 in gray. The resultant vector is shown following the same convention. (**e**) FM-amplitude as a function of tACS phase lag for 4 single-participant examples. Solid lines show true data and dashed lines the cosine fit. Only session one is shown.

The online version of this article includes the following figure supplement(s) for figure 3:

**Figure supplement 1.** Gap detection performance.

**Figure supplement 2.** FM-amplitude as a function of tACS phase lag.

## tACS does not overwrite FM-stimulus driven behavioral entrainment

First, we tested a strong version of a hypothesis about how tACS would affect behavior, which is that tACS would 'overwrite' neural entrainment to the FM stimulus. To do so, the phase lag between the FM-stimulus and tACS (Audio-tACS lag) was estimated for each trial. Gaps were grouped according to the FM-stimulus phase (9 bins) and the Audio-tACS lag (6 bins) and mean detection rates were computed for each FM-phase and Audio-tACS lag combination (*Figure 3a–b*; *Figure 3— figure supplement 1*). A cosine function was fitted to the mean detection rates as a function of FM-stimulus phase, separated by Audio-tACS lag, and the amplitude and optimal phase parameters

where estimated, similar to the sham condition. We hypothesized that tACS would interact with the FM-stimulus-driven neural entrainment and as such the behavioral signatures of entrainment – *FM-amplitude* and *FM-phase* – would vary as a function of Audio-tACS lag. First, if tACS overwrites entrainment to the FM-stimulus, we expect tACS phase to be more critical for gap detection than the phase of the auditory stimulus. As such, we anticipated a change in the FM-stimulus optimal phase as a function of Audio-tACS lag. Contrary to our expectations, in both sessions, optimal *FM-phase* was clustered across participants for all lag conditions around the same phase angle (V test with mean phase across lags as mean direction, all $V > 25.303$, $p < 2.016e\text{-}09$ *Figure 3d*). The same was true when separating by tACS montage (all $V > 11.729$, $p < 1.476e\text{-}04$). To further test the relevance of tACS phase for gap detection, we performed three mixed effect logistic regression models where single-trial gap detection performance was predicted by the FM-stimulus phase (model 1), plus the tACS phase at gap onset (model 2) and their interaction terms (model 3). Sham trials were excluded from this analysis. Before fitting the models, phase values were linearized by computing their sine and cosines. The best model was the one including only FM-stimulus phase (Δ AICc relative to model 2 = 6.20, *Supplementary file 1b*). This result, together with the lack of tACS effect on *FM-phase* indicates that tACS does not overwrite neural entrainment to the FM stimulus.

## tACS modulates the amplitude of the FM-stimulus driven behavioral entrainment

Our second hypothesis was that tACS could modulate the strength of entrainment to the FM stimulus, and as such, we expected *FM-amplitude* to be sinusoidally modulated by Audio-tACS lags: highest *FM-amplitude* should be observed for the optimal Audio-tACS lag since tACS would be aiding entrainment to the stimulus (constructive interference). In contrast, *FM-amplitude* was expected to be the smallest for tACS anti-phase to optimal lag: anti-phase tACS would disrupt entrainment to the FM stimulus since the electrical signal and the auditory rhythms are working against each other (destructive interference).

We wanted to test the effect of tACS lag on the FM-amplitude at the group level. However, it is already known that the optimal tACS lag for auditory perception often shows high inter-individual variability (*Riecke et al., 2015*; *Figure 3e*; *Figure 3—figure supplement 2*), which can affect the estimation of group-level tACS effects. To prevent inter-individual variability in optimal tACS lag from affecting the estimation of tACS effects at the group level, (1) individual optimal tACS lag (*tACS-phase*) was estimated for each participant and session using a cosine fit (Methods, *Figure 3e*) and (2) tACS lags were realigned by making the optimal *tACS-phase* correspond to phase zero and wrapping the remaining phases accordingly before doing group statistics (*Figure 4a*). Data from the optimal tACS lag and its opposite lag (corresponding trough) were excluded to avoid any artificial bias in estimating tACS effects induced by the alignment procedure (*Zoefel et al., 2019*). Data from the two tACS lags on either side of the optimal lag were averaged and used as an estimate of the FM-driven modulation in the optimal half of the tACS cycle (tACS(+)). Similarly, data from the two tACS lags on either side of the trough were averaged and used as an estimate of the FM-driven modulation in the suboptimal half of the tACS cycle (tACS(-)). A schematic of the calculation for obtaining tACS(+) and tACS(-) is shown in *Figure 4b*. tACS effects were tested using a mixed ANOVA with the repeated measures tACS lag condition (sham, tACS(+), tACS(-)) and session (1 and 2) and the between factor montage (*standard vs. individualized*). In addition, we included all two- and three-way interactions. Note that we chose to adopt this realignment procedure as done in previous studies (*Riecke et al., 2015*), instead of using the amplitude of the second cosine fit, because we were interested in investigating whether tACS could aid or disturb behavioral entrainment to the FM stimulus relative to the sham condition.

tACS did successfully modulate FM-stimulus driven behavioral entrainment (main effect of tACS lag condition: $F_{(2, 70)} = 26.202$, $p = 3.203e\text{-}09$, *Figure 4c*). Post-hoc *t-tests* showed that FM-driven modulation was higher for tACS(+) when compared to sham $t_{(36)} = 3.562$, $p = 0.003$ and to tACS(-) ($t_{(36)} = 9.501$, $p < 0.001$, all p-values are *Bonferroni* corrected, *Figure 4c*). In addition, FM-driven modulation was significant lower at tACS(-) when compared to sham ($t_{(36)} = -3.090$, $p = 0.012$, *Figure 4c*). No significant effect of session ($F_{(1,35)} = 0.083$, $p = 0.775$), montage ($F_{(1,35)} = 1.602$, $p = 0.214$) or interactions (montage*session $F_{(1,35)} < 0.001$, $p = 0.985$; montage*tACS lag $F_{(2,70)} = 0.282$, $p = 0.755$; tACS lag*session $F_{(2,70)} = 1.033$, $p = 0.361$; tACS lag*montage*session $F_{(2,70)} = 0.098$, $p = 0.907$) were

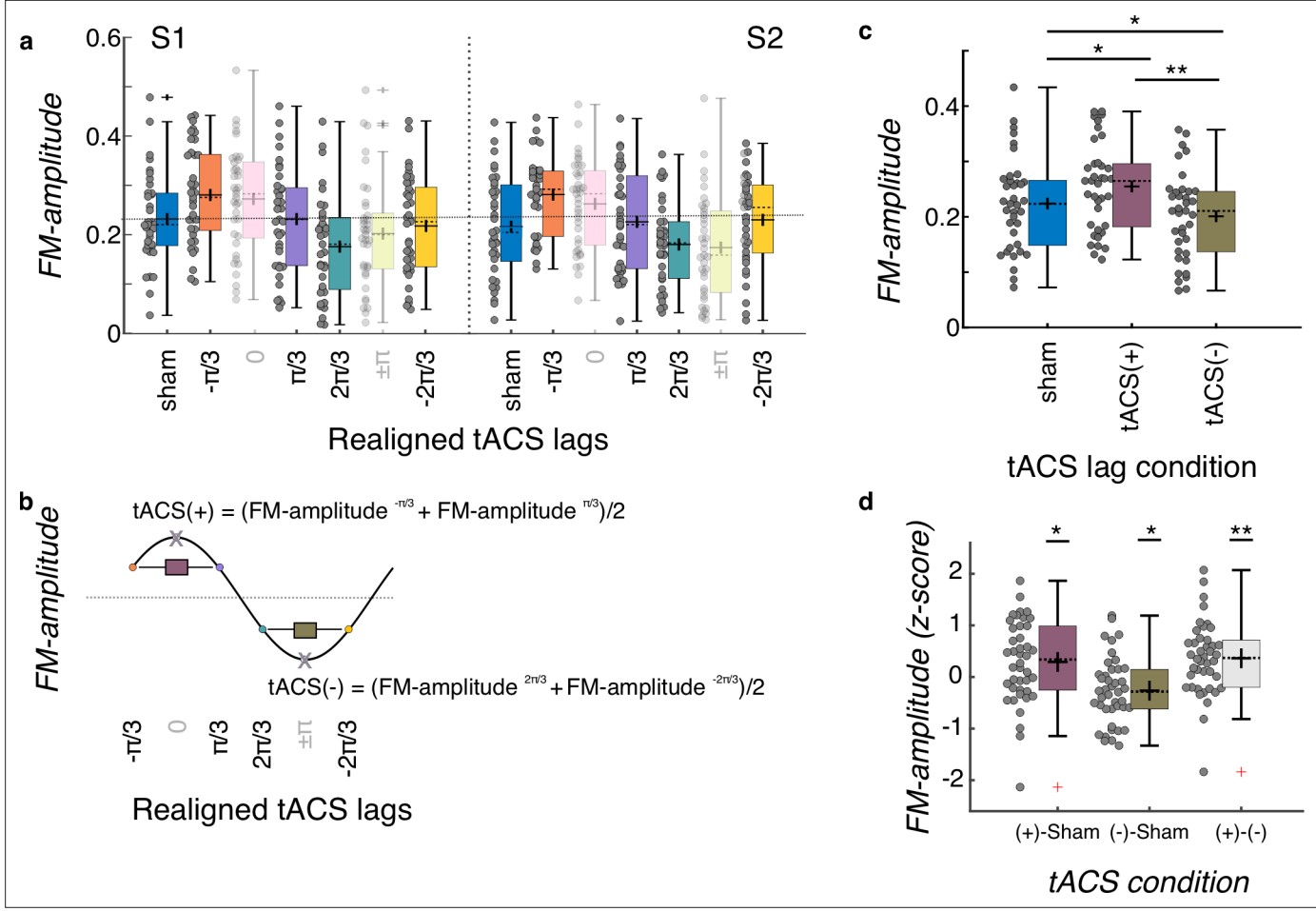

**Figure 4.** Group level tACS results. (**a**) Amplitude of the FM-stimulus driven behavioral modulation (*FM-amplitude*) as a function of the realigned tACS lag conditions. Zero lag corresponds to each individual optimal tACS lag (based on a cosine fit). N = 37. (**b**) *FM-amplitude* for optimal lag and the opposite lag (x letter on top of empty circle here, pink and green semitransparent bars in a) were removed from further analyses and estimates of *FM-amplitude* for the positive (tACS(+)) and negative (tACS(-)) tACS half cycles were obtained by averaging the individual FM-amplitude values from the two bins adjacent to the optimal lag and its corresponding trough, respectively. (**c**) FM-amplitude values estimated for sham, tACS(+), and tACS(-), as described in (**a, b**), averaged over sessions. *p = 0.05, **p < 0.001, post-hoc paired-samples t-tests, Bonferroni corrected. N = 37. (**d**) *FM-amplitude* difference between the 3 tACS conditions in c normalized (z-scores) to the permuted distributions. In all plots in the figure, each dot represents a single participant. Box plots show median (dashed vertical lines), mean (cross in the middle of the box), 25th and 75th percentiles (box edges) and extreme datapoints not considered outliers (+/–2.7σ and 99.3 percentiles, whiskers). Crosses represent outliers (more than 1.5 times the interquartile range away from the bottom or top of the box). N = 42. S1: session 1, S2: session 2. *p = 0.05, **p < 0.001, one-sample t-test, Holm-Bonferroni corrected.

The online version of this article includes the following figure supplement(s) for figure 4:

**Figure supplement 1.** tACS effects separated by tACS montage.

observed. In order to further investigate whether the observed tACS effect was significantly larger than chance and not an artifact of our analysis procedure (*Zoefel et al., 2019*), we created 1000 surrogate datasets per participant and session by permuting the tACS lag designation across trials. The same binning procedure, realignment, and cosine fits were applied to each surrogate dataset as for the original data. This yielded a surrogate distribution of tACS(+) and tACS(-) values for each participant and session. These values were averaged across sessions since the original analysis did not show a main effect of session. We then computed the difference between tACS(+) and sham, tACS(-) and sham, and tACS(+) and tACS(-), separately for the original and surrogate datasets. The obtained difference for the original data where then z-scored using the mean and standard deviation of the surrogate distribution. Note that in this case we used data of all 42 participants who had at least one valid session (37 participants with both sessions). Three one-sample t-tests were conducted to investigate whether the size of the tACS effect obtained in the original data was significantly larger than

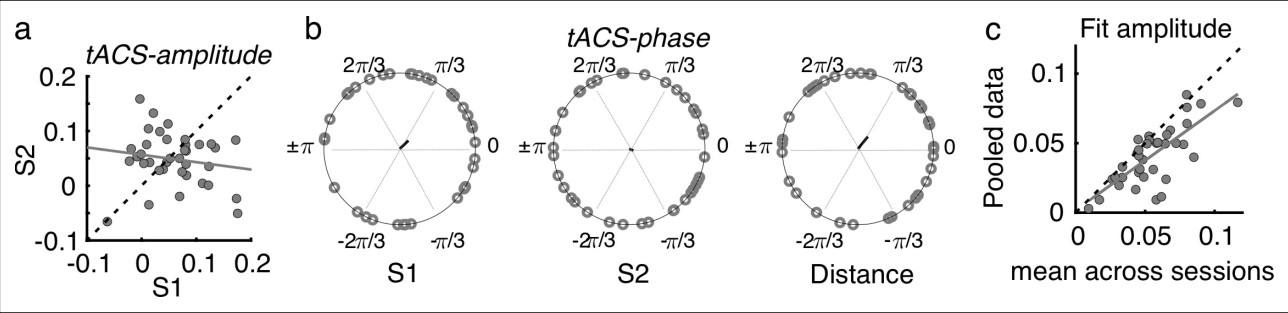

**Figure 5.** Effects of tACS on behavioral signatures of entrainment were not reliable over sessions. (**a**) Amplitude of the tACS effect (*tACS-amplitude*) for session 2 as a function of session 1. *tACS-amplitude* was computed as the difference between the *FM-amplitude* values at tACS (+) and tACS (-) in *Figure 4b*. Dashed line is the diagonal and gray solid line represents the regression line. Each dot represents a participant (**b**) Optimal tACS phase lag (*tACS-phase*) estimated from the cosine fits in (*Figure 3e*). The first two circular plots show optimal *tACS-phase* for each session while the third one shows the circular distance between sessions. The black line is the resultant vector. (**c**) The scatter plot shows the amplitude parameter obtained from fitting the cosine function to each session independently (as in **b**) and then averaged across sessions as a function of the fit amplitude obtained when fitting the cosine function to the data pooled across sessions. S1: session 1, S2: session 2.

The online version of this article includes the following figure supplement(s) for figure 5:

**Figure supplement 1.** Predicting inter-session difference in tACS-amplitude.

that obtained by chance (*Figure 4d*). This analysis showed that all z-scores were significantly higher than zero (all $t_{(41)} > 2.36$, $p < 0.05$, all p-values corrected for multiple comparisons using the *Holm-Bonferroni* method). Thus, tACS modulated behavioral signatures of neural entrainment to auditory rhythms, and critically both *amplified* and *attenuated* entrainment relative to sham depending on the phase relationship between the electrical and auditory stimulation. Such effect was not different either between session or montage group.

## Inter-individual variability and reliability of tACS effects

So far, we have evaluated whether tACS at different phase lags modulates behavioral signatures of entrainment: *FM-amplitude* and *FM-phase*. At the group level, we observed no modulation of optimal FM-stimulus phase (*FM-phase*) but a significant effect of tACS-lag on the FM-stimulus driven behavioral modulation (*FM-amplitude*). Next, we asked how reliable tACS effects are over time both in terms of the strength of tACS effects and the optimal tACS lag. This is critical to quantify if we want to plan tACS interventions using data from different sessions. For example, if optimal tACS lag constitutes a stable individual trait measure, it could be estimated in one session and used for applying tACS in following sessions to maximize treatment benefits.

To have a global measure of the magnitude of the tACS effect at the individual level, for each participant and session, the FM-driven modulation amplitude from the tACS negative half (tACS(-)) was subtracted from that during the tACS positive half (tACS(+)). This value was then interpreted as the amplitude of the tACS-induced behavioral modulation (*tACS-amplitude*, see Materials and methods, *Figure 5a*). To evaluate the reliability of the tACS induced modulation, *tACS-amplitude* values were compared and correlated between sessions. While no significant difference was observed between sessions, *tACS-amplitude* estimates did not correlate between sessions (session 1 vs session 2, $t_{(36)} = 0.806$, $p = 0.426$, Pearson's $r = -0.162$, $p = 0.337$, *Figure 5a*), suggesting a lack of reliability of the strength of tACS effects over time. No significant inter-session correlation was observed also when separating by montage group (all $|r| < 0.354$, $p > 0.179$).

Regarding optimal *tACS-phase*, as already mentioned in previous sections, we observed high inter-individual variability as the Rayleigh test did not indicate any significant deviation from uniformity in either session (session 1: $Z = 0.931$, $p = 0.397$; session 2: $Z = 0.133$, $p = 0.877$, *Figure 5b*). In addition, no significant correlation between sessions was observed (circular correlation, Rho = 0.237, $p = 0.137$). Moreover, the distribution of phase distances between sessions was not clustered around zero (V test with mean direction 0, $V = 4.838$, $p = 0.130$, *Figure 5b*). Same was true when separating by montage group (all $Z < 3.780$, $p > 0.122$, $|r| < 0.245$, $p > 0.272$). Thus, contrary to the reliability we observed for the optimal *FM-phase* for gap detection, optimal *tACS-phase* lags were not stable across sessions.

We anticipated that the lack of reliability of optimal tACS phase could be influenced by a poor esti-mation of each single-session optimal phase lag due to a low number of trials per condition (FM-stim-ulus phase x Audio-tACS lag). If so, we hypothesized that the estimation of tACS effects should be strengthened by pooling data across sessions, as this would increase number of trials by a factor of 2. In this case, we expect *tACS-amplitude* to be equal or higher when estimated with the pooled data than when estimated for each separate session. However, we observed that *tACS-amplitude* was *smaller* when calculated with the pooled data than when averaging the *tACS-amplitude* across sessions ($t_{(36)}$ = –2.452, p = 0.019, *Figure 5c*). We interpret the smaller *tACS-amplitude* as an indication of tACS effect cancellation due to inter-session variability. In other words, this result suggests that the lack of reliability in *tACS-phase* cannot be explained by the number of trials used for its estimation.

To better understand what factors might be influencing inter-session variability in tACS effects, we estimated multiple linear models predicting inter-session difference in *tACS-amplitude* (session 1- session 2) and absolute circular distance in *tACS-phase* between sessions from a different combi-nations of regressors: (1) age, (2) gender, (3) number of days passed between sessions, ΔDays, (4) absolute time of the day difference in minutes, ΔMinutes, (5) tACS-montage, and (6) inter-session difference in gap size threshold. The winning model (*F*-statistic vs. constant model: 5.72, p = 0.007, $R^2$ = 0.257, *Supplementary file 1c*) showed that inter-session difference in *tACS-amplitude* was signifi-cantly predicted by the number of days passed between sessions (ΔDays *ß* = –0.383, *t* = –2.55, p = 0.016) and the absolute time of the day difference (ΔMinutes *ß* = 0.350, *t* = 2.333, p = 0.026). The visualization of these effects suggested that, compared to the first session, *tACS-amplitude* decreased in session 2 for participants with longer intervals passed between sessions and for participants tested at the same time of the day (*Figure 5—figure supplement 1*). This model was selected because it had the smallest Akaike's information criterion (corrected for small samples), AICc. Moreover, the likeli-hood ratio test showed no evidence for choosing the more complex unrestricted model (*stat* = 2.411, p = 0.121). Following the same selection criteria, the winning model predicting inter-session variability in tACS-phase, included only the factor gender (*Supplementary file 1d*). However, this model was not significant in and of itself when compared to a constant model (*F*-statistic vs. constant model: 3.05, p = 0.09, $R^2$ = 0.082). Therefore, contrary to *tACS-amplitude*, the lack of reliability in *tACS-phase* cannot be explained by the interval passed between sessions or time of the day, but rather may be a more fundamental feature of intra-individual variability that should potentially influence the design of tACS experiments, in particular when conducted across multiple sessions.

## Inter-individual variability in the simulated E-field predicts tACS effects

So far, our results demonstrate that FM-stimulus driven behavioral modulation of gap detection (*FM-amplitude*) was significantly affected by the phase lag between the FM-stimulus and the tACS signal (*Audio-tACS lag*). However, the strength of the tACS effect was variable across participants. A natural question for us was what factors predict individual variability in the magnitude of the tACS effect. Specifically, we sought to explain variability in tACS effects from nine different variables including the seven parameters describing the optimality of the E-field simulated for each participant (*Figure 2b*); the strength of the BOLD signal response in the ROI to the auditory stimulus and the FM-driven behavioral modulation during sham stimulation. *tACS-amplitude* estimates were averaged across sessions since session did not significantly affect *FM-amplitude* (*Figure 5a*). Participants missing the fMRI session were excluded and the analysis was performed with data from 34 participants (indi-vidualized montage N = 16, standard montage N = 18). Different linear models were fitted to the data aiming to predict tACS modulation strength (averaged *tACS-amplitude*) from different combinations of regressors (see Materials and methods). The winning model (*F*-statistic vs. constant model: 3.41, p = 0.016, $R^2$ = 0.378, *Figure 6*, *Supplementary file 1e*) included the main effects: normal E-field (*ß* = –0.112, *t* = –0.656, p = 0.517), E-field focality (*ß* = 0.369, *t* = 1.969, p = 0.059), average distance between the E-field peaks and the center of the target functional ROIs for each hemisphere (*Dist-2Peak; ß* = –0.054, *t* = –0.328, p = 0.746), and the interaction terms: normal E-field: focality (*ß* = 0.735, *t* = 3.260, p = 0.003) and normal E-field:*Dist2Peak* (*ß* = 0.548, *t* = 2.804, p = 0.009). Note however that only the interaction terms were significant.

Visualization of the interaction terms showed that tACS effects were positively predicted by the normal E-field for participants with more focal E-fields that were closer to the functional target ROIs (*Figure 6*). However, in participants where the induced E-field was less focal and farther from the

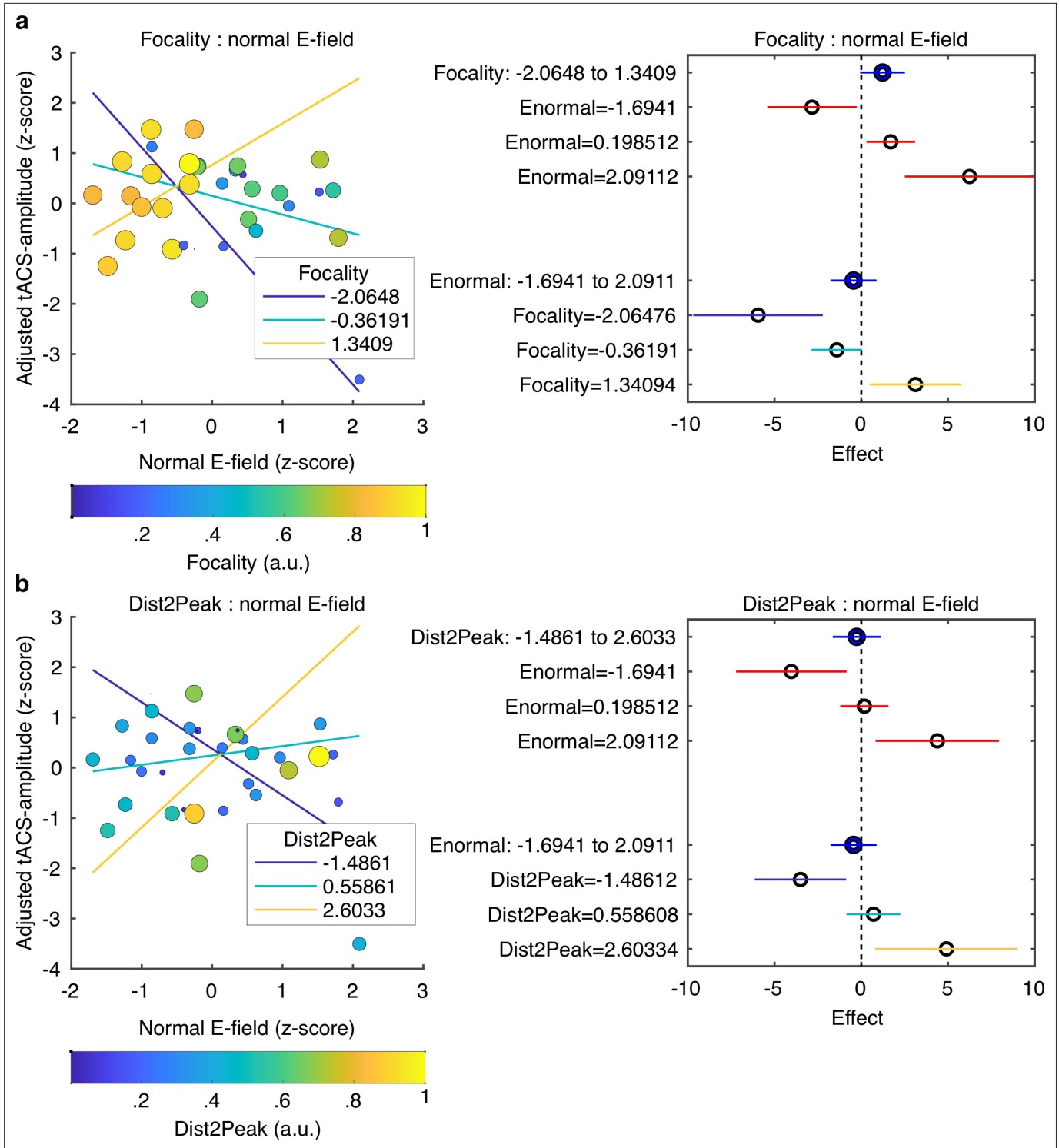

**Figure 6.** Predicting tACS effects from electric field simulation. (**a**) Interaction between the normal E-field and the field focality. Scatter plot on the left shows tACS effects (*tACS-amplitude*) as a function of the normal E-field. Each dot represents a different subject. Dot color and size represent the normalized field focality in arbitrary units. Higher values correspond to more focal electric fields. Solid lines represent the predicted adjusted response for the *tACS-amplitude* as a function of the normal E-field for three fixed values of focality. The plot in the right shows the main effects (blue) of focality and normal E-field and the conditional effect of each predictor given a specific value of the other (red). Horizontal lines through the effect values indicate their 95% confidence intervals. (**b**) Interaction between the normal E-field and the distance between the peak of the E-field and the target ROIs (*Dist2Peak*). Similar to (**a**), scatter plot on the left shows *tACS-amplitude* as a function of the normal E-field but dots color and size now represent the normalized *Dist2Peak*. Higher values correspond to shorter distance. Solid lines represent the predicted adjusted response for the *tACS-amplitude* as a

*Figure 6 continued on next page*

*Figure 6 continued*

function of the normal E-field for three fixed values of *Dist2Peak*. The plot in the right shows the main effects (blue) of *Dist2Peak* and normal E-field and the conditional effect of each predictor given a specific value of the other (red). Horizontal lines through the effect values indicate their 95% confidence intervals. Colors in the bottom for the Focality and *Dist2Peak* levels correspond to the same color code in the upper plots.

functional target, the normal E-field negatively predicted tACS effects. The latter can be seen in the plots in *Figure 6*, where the adjusted tACS effects are shown as a function of the normal E-field at three different fixed values of focality (*Figure 6a*) and *Dist2Peak* (*Figure 6b*). Thus, when the electric field accurately reaches its intended target, high field strength increases behavioral consequences of tACS, but when the electric field misses its mark, high field strength is actually behaviorally detrimental.

## Can we reduce inter-individual variability in tACS effects with individually optimized tACS montages?

When testing the effect of tACS on the amplitude of behavioral entrainment to the FM stimulus (*FM-amplitude*), we did not observe a significant effect of tACS montage (section *tACS modulates the amplitude of the FM-stimulus driven behavioral entrainment*). In other words, the specific tACS montage did not have a big impact on the size of the tACS effects overall (see Discussion). However, since testing the impact of individualizing the tACS montage on behavioral effects was one of the main aims of the current project, we nonetheless tried to investigate this further.

In the individualized montage group, we optimized each listener's montage by taking into account individual anatomical and functional data. One possible consequence of such approach could be a lower inter-individual variability in tACS effects, rather than an overall increase in effect size. To test this hypothesis, participants were grouped according to the electrode montage and individual *tACS-amplitude* values (see previous section) were averaged across sessions. A variance test was performed on the average *tACS-amplitude* to test whether inter-individual variability was different between montage groups. For this analysis, all participants who completed both sessions were included (individualized N = 16, standard N = 21). No significant difference was observed in terms of the variance between groups ($F_{(20, 15)} = 1.441$, p = 0.475).

We then took a step back and compared the group variances separated by session, and by tACS condition (tACS(+), and tACS(-), *Figure 4—figure supplement 1*). Visual inspection of the plot suggested that inter-individual variability was smaller for the individualized montage group than for the standard montage group. To quantify this, we computed the coefficient of variation (cv) and interquartile range (iqr) for each group (session 1, individualized: tACS(+) cv = 0.3567, iqr = 0.1375, tACS(-) cv = 0.4848, iqr = 0.1480; standard: tACS(+) cv = 0.3757, iqr = 0.1684, tACS(-) cv = 0.4792, iqr = 0.1497; session 2, individualized: tACS(+) cv = 0.2531, iqr = 0.1086, tACS(-) cv = 0.2201, iqr = 0.0659; standard: tACS(+) cv = 0.3917, iqr = 0.1704, tACS(-) cv = 0.5174, iqr = 0.1736). However, the variance test was significant only for tACS(-) in session 2 ($F_{(20,15)} = 4.31$, p = 0.023, *Bonferroni* corrected for 4 comparisons). No significant differences in group variance were observed for the other conditions (all $F_{(20,15)} < 1.924$, p > 0.20, uncorrected). This result is suggestive, but larger sample sizes would be necessary to conduct a principled variance comparison between groups. At this stage, more evidence is needed to prove the superiority of individually optimized tACS montages for reducing inter-individual variability in tACS effects.

## Discussion

In the present study, we used tACS to modulate neural entrainment to FM sounds. We tested the relevance of tACS phase lag relative to the sound rhythm and tACS montage for modulating behavioral signatures of entrainment. Our main findings show that tACS (1) does not modify the optimal FM-stimulus phase for gap detection (*FM-phase*), but (2) does modulate the amplitude of the FM-stimulus induced behavioral modulation (*FM-amplitude*) in a tACS-lag specific manner; (3) Neither the optimal tACS lag (*tACS-phase*), nor the magnitude of the phasic tACS effect (*tACS-amplitude*) were reliable across sessions; and (4) individual variability in tACS effect size was partially explained by two interactions: between the normal component of the E-field and the field focality, and between the normal component of the E-field and the distance between the peak of the electric field and the functional target ROIs. Overall, these interactions indicate that when the electric field reaches its intended

target, high field strength leads to stronger behavioral effects, but when the electric field is stronger in non-target brain areas, higher field strength actually reduces the behavioral effect size.

## tACS modifies the amplitude, but not the phase, of FM-stimulus induced behavioral modulation

Participants detected near-threshold silent gaps embedded in a 2 Hz FM noise stimulus. Using this paradigm, it has been repeatedly shown that EEG activity entrains to the FM rhythm and that detection of silent gaps presented in the sound is modulated by the FM-stimulus phase into which the gap falls (*Cabral-Calderin and Henry, 2022*; *Henry and Obleser, 2012*; *Henry et al., 2014*; *Bauer et al., 2018*). Moreover, we have recently shown that both effects are reliable across participants and over time (*Cabral-Calderin and Henry, 2022*). Our first hypothesis was that tACS could overwrite neural entrainment to the FM stimulus (i.e. neural oscillations phase-lock to the tACS signal instead of the FM-stimulus) and as such, the optimal FM-stimulus phase would vary as a function of tACS lag. Contrary to our prediction, results showed that in both sessions, individual optimal FM phases for gap perception were always clustered around the same phase angle, regardless of the tACS lag. We hypothesize that a reason for this could be the already strong acoustic modulation of the stimuli that we used here, which had a modulation depth of 67% peak-to-center. Phasic effects of tACS on auditory perception have previously been shown mostly in the context of weakly modulated background sounds (*Riecke et al., 2015*) or for example when tACS was applied with the shape of an amplitude envelope that had been acoustically removed from a speech signal (*Zoefel et al., 2018*; *Zoefel et al., 2020*). We hypothesize that entrainment to the clear rhythm of the FM stimulus might have been too strong for tACS to overwrite. It has already been suggested, at least for tACS applied at frequencies matching the neural alpha band, that tACS is most effective at modulating oscillatory activity at the intended frequency when its power is not too high (*Neuling et al., 2013*). Along the same line, using single unit recordings it has recently been proposed that tACS competes with endogenous neural oscillations: tACS imposes its own rhythm on spiking activity when tACS strength is stronger than the endogenous oscillations but it decreases rhythmic spiking when tACS strength is weaker that the endogenous oscillations (*Krause et al., 2022*). Thus, we would predict that tACS may be more likely to shift the phase of neural entrainment when entrainment is weaker to start with, either because of weaker stimulus modulation or in listeners lacking young, healthy auditory systems (*Rufener et al., 2016*). Alternatively, our results could simply be interpreted as a clear superiority of the auditory stimulus for entrainment. In other words, sensory entrainment might just be stronger than tACS entrainment in this case where the stimulus rhythm was strong and salient. It would be interesting to further test whether this superiority of sensory entrainment applies to all sensory modalities or if there is a particular advantage for auditory stimuli when they compete with electrical stimulation. However, answering this question was beyond the scope of our study and needs further investigations with more appropriate paradigms.

Our second hypothesis was that, while tACS might not be able to overwrite entrainment to a strongly modulated FM stimulus in healthy young adults, it could still modulate the amplitude of the FM-driven entrainment. In other words, we anticipated that tACS would interfere with neural entrainment to the FM-stimulus depending on their relative phase lag: optimal tACS lag would aid neural entrainment (increase amplitude of FM-driven modulation) as in constructive interference, while the opposite lag would be detrimental (decrease amplitude of FM-driven modulation) since both the FM stimulus and the tACS signals would try to entrain neural activity with antiphase relationships, as in destructive interference. Our results showed that the amplitude of the FM-stimulus driven modulation (*FM-amplitude*) was sinusoidally modulated based on the relative phase between the stimulus rhythm and the electrical stimulation. Specifically, *FM-amplitude* was stronger for the phase lags around the realigned optimal tACS lag and weaker for the tACS lags opposite the optimal lag. The fact that FM-stimulus-driven modulation of gap performance was affected by tACS in a phase-specific manner implicitly suggests that neural entrainment to auditory stimuli is causally linked to auditory perception and that tACS is an effective tool for modulating both. Based on the association between neural entrainment and auditory perception (*Peelle and Davis, 2012*; *Zion Golumbic et al., 2013*; *Horton et al., 2013*; *Doelling et al., 2014*; *Horton et al., 2014*; *Doelling and Poeppel, 2015*; *Nozaradan et al., 2016*; *Brodbeck et al., 2020*), we anticipate that making entrainment stronger with tACS should lead to better cocktail party performance or better prediction/adaptation in a musical context.

Note that we did not control for the skin sensation induced by 2 Hz tACS in this experiment. Participants rated the strength of the perceived stimulation after each run. However, this information was used only to assess the safety and tolerability of the stimulation protocol. It is in principle possible that skin sensation would depend on tACS phase itself. However, in this study, we report effects that depend on the relationship between tACS-phase and FM-stimulus phase, which changed from trial to trial as the starting phase of the FM-stimulus was randomized across trials. We have no reason to expect the skin sensation to change with the tACS-audio lag and therefore do not consider this to be a confound in our data. We propose that by applying tACS at the right lag relative to auditory rhythms, we can aid how the brain synchronizes to the sounds and in turn modulate behavior. Since optimal tACS phase was variable across participants and sessions, this approach would require closed-loop protocols where the optimal tACS lag is estimated online (see next section). In addition, we anticipate that such an approach might work even better when entrainment to the auditory stimulus is not perfect, for example when the driving stimulus is weaker (weaker modulation) or masked by noise.

## tACS effects were variable across sessions

The reliability of tACS effects at the individual level is not fully understood. While inter-individual variability in response to electrical stimulation has been well documented (*Kasten et al., 2019*; *Cabral-Calderin et al., 2016b*; *Zanto et al., 2021*; *Li et al., 2015*), most studies test tACS in only one session, and when multiple sessions are conducted, a different tACS condition is implemented in each session (*Cabral-Calderin et al., 2016b*; *Antal et al., 2008*). Here, we tested participants under the same auditory and tACS conditions in two different sessions. This approach allowed us to test the stability of tACS effects across sessions. While no significant difference between sessions was observed for the amplitude of tACS effects (*tACS-amplitude*), tACS effects did not correlate between sessions. Moreover, no significant correlation was observed between optimal tACS phase lags (*tACS-phase*) across sessions, and the circular distance between sessions was not clustered.

In a previous study using stimuli with exactly the same parameters as in the current study, we showed neural entrainment to FM-sounds has a reliable phase lag over sessions (*Cabral-Calderin and Henry, 2022*), so any lack of reliability in optimal tACS lag cannot be explained by inconsistency in FM-stimulus-brain synchrony. One possible explanation for the lack of reliability of tACS effects could be differences in the placement of the tACS electrodes across sessions. We took special care to precisely and consistently place electrodes across sessions, thus we feel confident that our electrode placement did not significantly differ between sessions. Nevertheless, even subtle differences could induce changes in the precise angle of the current dipole induced in the auditory cortex or slightly affect the optimal spatial targeting of the field. Such subtle changes in electrode placement could lead to different frequency-tuned neurons receiving the strongest stimulation in every session. However, we encourage the reader to be careful with such interpretation. In this experiment, we used frequency modulated sounds. At the level of the auditory pathway, frequency modulation can be translated into amplitude-modulation of different frequency-tuned neurons: neuronal populations will be activated on and off depending on whether their preferred frequency is included in the FM stimulus. Within a given FM stimulus, different tACS lags will correspond to different center frequencies in the FM stimulus: highest current strength will correspond to a specific frequency. Note however that we randomized the stimulus center carrier frequency from trial to trial. Therefore, there is no one-to-one mapping between tACS phase and FM-stimulus carrier frequency. This idea would require further testing and systematic investigation into whether small changes in electrode placement changes optimal tACS lag and to what extent this is affected by the stimulus carrier frequency. We followed the standard procedure in the field for placing the tACS electrodes. If such subtle differences in electrode placement can already destroy inter-session reliability, there's really no hope for any human to place them more precisely than that, so we will need to create new approaches in the field (e.g. similar to the neuronavigation systems used for TMS) to guarantee even more precise electrode placements than using the standard EEG 10–20 system. The idea that differences in optimal tACS lag can be explained by different frequency-tuned neurons being activated in each session relies on the spectral representation of FM sounds in the auditory pathway. However, neural processing of FM seems to involved not only frequency encoding but also direction and rate encoding: neurons selectively responding to FM direction or rate (*Tabas and von Kriegstein, 2021*). The latter makes the understanding of variability in optimal tACS phase even more complex.

Interestingly, our results suggest that the inter-session interval and the difference in time of the day are two factors influencing the inter-session variability in the strength of tACS effects. Such finding suggests that inter-session variability in tACS effects cannot be exclusively explained but inconsistency in the electrode placement.

Several physiological factors could be behind the influence of inter-session intervals on inter-session reliability of tACS effects. For example, hormone changes in women related to the menstrual cycle have been shown to alter cortical excitability (*Inghilleri et al., 2004*). In the latter study, repetitive TMS had a facilitatory effects on motor evoked potentials on day 14 of the menstrual cycle but not on day 1. We did not control for the phase of the menstrual cycle of our female participants but our inter-session intervals suggest that female participants were at different phases in each session. However, if hormonal factors would explain the lack of inter-session reliability of tACS effects, we would expect male participants to be more reliable than female ones. However, we did not observe any effect of gender when investigating inter-session reliability in our models. Another factor to consider when interpreting inter-session variability of tACS effects is plasticity induced by the first session. In session 1, participants are less experienced with the task and as a result perceptual learning slopes might be bigger and differently interfere with tACS effects. In fact, within the same day, the effects of anodal transcranial direct current stimulation in a visual discrimination task seems to differ between blocks, being greatest at the beginning and not different from sham over time (*Fertonani et al., 2011*). In principle, adaptation to the electrical current in the auditory regions could change the weight of different input neuronal connections and as such change the optimal phase for tACS to affect neural entrainment to the auditory stimulus. Alternatively, another plausible explanation is that participants could have changed the strategy to perform the task in each session (*Cabral-Calderin and Henry, 2022*) leading to different cognitive processes being relevant in the different sessions and hence inducing more variable tACS effects.

The lack of test-retest reliability on tACS effects is an especially surprising result from the current study which requires further investigation. The implication of this result is it may not be possible to always estimate optimal tACS phase lag in a session using magneto- or electroencephalography and then apply it in a second session (*van Bree et al., 2021*), but closed-loop protocols where optimal tACS lag is constantly monitored during online tACS may be necessary. In a previous study (*van Bree et al., 2021*), the authors where able to predict optimal tACS phase for speech comprehension from EEG data. Note however, that this was possible only when the speech signal was presented after the tACS signal (no competition between the tACS and the sensory signal). In fact, when the auditory stimulus was presented during tACS, the authors showed no tACS-induced modulation of performance and optimal tACS phase could not be predicted from EEG data. Moreover, optimal tACS phase during stimulation did not correspond with estimated optimal phase in the post-stimulation period. Taken together, results from both studies suggest that optimal tACS phase might be stable only in the post-stimulation period, when tACS and the sensory signal do not compete with each other, and only endogenous "echoes" of entrained neural oscillations are expected to occur. In our paradigm, we only tested tACS effects while a strongly modulated FM stimulus at the tACS rate was presented. Competition between the electrical and acoustic signals for entrainment of endogenous oscillations, together with evoked responses to each signal modality and the electric signal per se (strong current vs. none) adds more variability in the optimal tACS phase. Other differences between the study designs include the sensory stimulus type (intelligible speech vs. FM sounds), which might involve different neural processes, and the tACS duration (burst of 3–6 s tACS with vs. 11 minutes tACS continuous with 20 s FM). Stability of tACS phases might depend as well on the stimulation length. In addition, in *van Bree et al., 2021*, tACS was applied only in one session. It remains to be seen whether optimal tACS phase for perception remains stable or if it changes over time due plasticity-induced changes as discussed above. In the latter case, optimal tACS phase would be predicted from EEG activity in the first tACS session but not in subsequent ones. In any case, the inconsistency of optimal tACS lags across sessions observed in our study suggests that individual variability in optimal tACS lags to modulate entrainment to rhythmic auditory stimuli does not arise exclusively from anatomical variability but other changes in functional brain activity might be important to consider. More research is needed to fully understand optimal tACS phases for perception.

## tACS effects can be predicted from the simulated electric field

Several studies have already shown that inter-individual variability in tACS effects can be partially explained by inter-individual variability in the strength of the simulated electric field as well as the spatial overlap between this and the targeted functional area (*Kasten et al., 2019*; *Zanto et al., 2021*; *Preisig and Hervais-Adelman, 2022*). Here, we used a model-comparison approach to determine the parameters that best explained individual variation in tACS effects; we started with E-field strength, E-field focality, the correspondence between the simulated E-field and the target functional regions (correlation between E-field map and BOLD map, Euclidian distance between E-field and BOLD peaks), baseline factors (FM-driven modulation amplitude at sham and the strength of the BOLD signal at the regions of interest) and the tACS montage. The best model showed that tACS effects were best explained by the interaction between the normal component of the E-field and the field focality, and the interaction between the normal E-field and the distance between the E-field peak and the peak of the functional activation map. The explanatory power of these interaction terms demonstrates that not only the field strength is important, but also how focal it is and how optimally it targets the intended brain region. For more focal electric fields targeting the correct brain regions, higher normal E-fields are beneficial. However, higher normal E-fields can be detrimental when the field is too broad and its peak farther away from the target ROI. This potentially relates to the fact that stronger electric fields will be applied to non-relevant brain areas with unintended consequences, potentially reducing the efficacy of tACS to modulate performance in the target task.

## Are individually optimized tACS montage better?

In the first part of this study, we simulated Electric field strength using 3 different tACS montages. The first two montages were taken from the literature (*van Bree et al., 2021*; *Baltus et al., 2018*) and were the same for all subjects, while the third montage was individually optimized to target individual functional ROIs using an fMRI-task localizer. Before running any simulations, we had already decided a priori to use only the individualized and standard montages for our main behavioral experiment, motivated by the fact that the 4 electrodes arrangement allows for directing the E-field in a desired orientation (*Baltus et al., 2018*). Directing the E-field in a desired orientation is not possible using the ring electrodes configuration. Ring-electrode montages will always have a focus on superficial brain areas under the center electrode, and the E-field direction in the focus will be normal to the brain surface. As already described, we did not observe any difference between the individual and the group montages in terms of spatial overlap to our target ROI. While the electric field was strongest for the standard montage compared to the individualized montage, the latter induced a more focal field compared to the standard montage, therefore each montage could be good in different ways.

By using the individualized montage, we aimed to go a step past previous studies, which have shown that inter-individual variability in the E-field predicts tACS effects. Here, we asked whether tACS effects could be strengthened if the montage is optimized to each participant's anatomy and functional map, and if such approach would also reduce inter-individual variability. Compared to the standard montage, we did not observe any significant different between the overall tACS effects. We did observe a trend for decreased inter-individual variability for the individualized montage group, although this was only statistically significant for session 2 and for the amplitude of the FM-driven modulation at the non-optimal half of the tACS lags (tACS(-)).

Several points need to be considered when interpreting these results, (1) the standard montage was taken from a study where it was already optimized using anatomical and functional data from a single participant (*Baltus et al., 2018*). Therefore, our findings speak not to whether optimizing tACS montages is beneficial in general, but whether spatially optimizing the montage to each participant adds extra advantages. It could well be that the standard montage was already optimal for several participants with more standard brains. In fact, our individualized montage was spatially not so different from the standard one. (2) In the current experiment, we only tested healthy young participants. We anticipate that taking into account inter-individual anatomical and functional variability will be more relevant in clinical settings or in aging research, where such variability is more evident. (3) We did not optimize the tACS protocol in terms of the applied current strength. We used a fixed value because we were limited by the size of the electrodes, and going higher than that value would make it uncomfortable for the participant. Combining spatial optimization with current-strength optimization could lead to different outcomes than what we observed here. (4) The optimization procedure

here was performed with the constraints of achieving the same E-field strength in both target ROIs, inducing a relatively balanced field in both hemispheres and allowing only for four electrodes, that is two per hemisphere. Different optimization results might be obtained when optimization for more stimulation channels and allowing for a quite liberal amount of total current.

As discussed above, our results showed that the strength of tACS effects was predicted by the interactions between the strength of the inward electric field with the field focality and the distance between the peak of the E-field and the target ROI. Since such information was not available before-hand, it was not considered in our optimization approach. Our montage was optimized to spatially overlap as much as possible with our functional target, which as expected turned out to be of relevance in the analysis. While we did not optimize for focality, the individualized montage was more focal than the standard montage, another factor contributing to tACS effects. However, we did not optimize for the strength of the inward electric field, since we were limited by how much current strength we could apply and aimed at inducing a symmetrical electric field in the brain. The standard montage in fact induced higher field strengths in the brain than our individualized montage. Future studies could try to systematically optimize for the 3 factors and test the impact of such procedures at the behavioral level. Our results suggest that in our current settings, although individualizing the tACS montage did not make a difference in terms of overall tACS effects, it could be a way to decrease inter-individual variability; given the highly variable nature of tACS effects in the literature, this may be desirable under some circumstances. However, our study was not designed to directly test the effect of tACS on sample variance, and more data would be needed to support this observation. To our knowledge, studies answering this question are missing in the tACS literature.

## Summary and future directions

In summary, our results showed that while tACS did not affect the optimal FM-stimulus phase for gap detection, it did modulate the amplitude of the stimulus-induced behavioral modulation, a behavioral signature of the strength of neural entrainment. Both the optimal tACS lag and the amplitude of tACS effect were variable across participants and sessions, pointing towards the potential usefulness of closed-loop stimulation protocols, where optimal lags are evaluated online during stimulation. tACS effects were predicted by the interaction between the normal component of the E-field and the field focality and the interaction between the normal E-field and the distance between the peak of the E-field and the peak of the target ROI. These results suggest that the strength of the electric field is more beneficial when the E-field is closest to the target ROI and more focal, while stronger normal E-field in less focal and more distant E-fields can be detrimental. While no significant effect of tACS montage was observed in the overall tACS effects, the data hint at lower inter-individual variability in the group with the individualized montaged compared to the standard montage.

Our results advance our understanding of the use of tACS for modulating entrainment to auditory rhythms and put us a step forward on the way to personalized brain stimulation. Likewise, our study opens many questions for future research: (1) what other factors underlie the inter-session variability of tACS effects, (2) what factors determine the optimal tACS lag beyond brain anatomy, (3) how variable are optimal tACS lags within an individual depending on the sensory stimulation condition or targeted brain region, (4) what other factors should we consider when optimizing brain stimulation montages beyond spatial targeting, (5) what is the impact of our optimization approach when applying tACS in clinical or older populations? While future studies replicating our findings are needed, two specific recommendations for future research can be derived from our results: (1) Spatially optimizing the tACS montage can be used when aiming at decreasing inter-individual variability of tACS effects; (2) tACS montages should be optimized not only for the electric field strength, but for the strength of its normal component, the field focality and its spatial overlap with the target ROI.

## Materials and methods

### Participants

Forty-two healthy participants took part in the study (18 females, mean age = 27.3; SD = 5.8). All participants took part in at least one tACS session and 39 participants took part in two tACS sessions separated by 3–301 days (median: 7 days). Variability in inter-session intervals was due to multiple lock-downs during the corona pandemic. Data from session one from one participant had to be excluded

due to technical problems with the tACS signal recording. From the initial sample, three individuals did not participate in the fMRI session due to MRI contraindications (metal implants). All participants self-reported normal-hearing and normal or corrected-to-normal vision. At the time of the experiment no participant was taking medication for any neurological or psychiatric disorder.

Participants received financial compensation for their participation in the study. Written informed consent was obtained from all participants. The procedure was approved by the Ethics Council of the Max Planck Society (registration *Saturnino et al., 2019*_27) and in accordance with the declaration of Helsinki.

## Auditory stimuli

Auditory stimuli were generated by MATLAB software at a sampling rate of 44.1 kHz. Stimuli were 20 s long complex tones frequency modulated at a rate of 2 Hz and with a center-to-peak depth of 67% of the center frequency (*Figure 1a*). The center frequency for the complex carrier signals was randomly chosen for each stimulus within the range of 1000–1400 Hz. The complex carrier comprised 30 components sampled from a uniform distribution with a 500 Hz range. The amplitude of each component was scaled linearly based on its inverse distance from the center frequency; that is, the center frequency itself was the highest-amplitude component, and component amplitudes decreased with increasing distance from the center frequency. The onset phase of the stimulus was randomized from trial to trial, taking on one of eight values (0, π/4, π/2, 3π/4, π, 5π/4, 3π/2, 7π/4), with the constraint that each trial would always start with a phase different from its predecessor. All stimuli were rms-amplitude normalized. Each 20 s stimulus contained 3–5 silent gaps (gap onset and offset were gated with 3 ms half-cosine ramps). Each gap was chosen to be centered in 1 of 9 equally spaced phase bins into which each single cycle of the frequency modulation was divided. No gaps were presented either in the first or the last second of the stimulus. A minimum of 1.5 s separated consecutive gaps.

## Procedure: Magnetic resonance imaging (MRI) session

### Data acquisition

Thirty-nine participants from the initial sample took part in one session where structural and functional MRI data were collected. All images were acquired using a 3 Tesla Magnetom Prisma scanner (Siemens Healthcare, Erlangen, Germany) with a 32-channel head coil with mirror system. A three-dimensional (3D) magnetization prepared rapid gradient echo (MPRAGE) T1-weighted scan (T1w; repetition time (TR) = 2s; echo time (TE) = 2.12ms; flip angle (FA) = 8°; 192 sagittal slices; matrix = 256 × 256; field of view (FOV) = 256 × 256 × 192mm3; voxel size = 1 mm³; bandwidth = 210Hz; selective water excitation; integrated parallel acquisition technique GRAPPA: factor 2) and a 3D turbo spin-echo (TSE) T2-weighted scan (T2w; TR = 1500ms; TE = 356ms; FA = 120°; 192 sagittal slices; matrix = 256 × 256; FOV = 256 × 256 × 192 mm3; voxel size = 1 mm³; bandwidth = 331Hz) were performed. All functional data were acquired using the 2D multiband gradient-echo echo planar imaging sequence (EPI) with T2*-weighting (TR = 1s, TE = 30ms, FA = 60°, 48 slices of 2mm thickness, 50% distance factor, in-plane resolution of 3×3 x 2 mm, multiband acceleration factor 4). A total of 590 whole-brain volumes were acquired in each functional run. Three functional (f)MRI runs were collected. To correct for field distortion in the functional images, a 5-volumes additional reference sequence was acquired after the first functional run with reversed phase-encoding direction resulting in images with distortions going in opposite directions to that of the main run.

### Auditory stimulation in the MRI scanner

During each functional run, participants listened to the FM stimulus and detected the silent gaps. Auditory stimuli were presented using an MR-compatible noise-cancelling headphones (OptoAcoustics OptoActive II TM ANC). These headphones reduced the EPI-gradient noise by about 20–30 dB. Prior to the main task, individual sound level and gap duration thresholds were estimated using the same procedure described for the main tACS sessions. To consider the influence of scanner noise during thresholds estimation, the same functional sequence as in the main task was run during the threshold tasks but fMRI data were not further analyzed. For the main functional runs, participants performed a version of the same task presented during the tACS sessions. Each 20 s FM-stimulus presentation was interleaved with a 20 s (+–1 s) rest period during which only a fixation cross was

presented in the screen. In total, participants listened to 42 FM-stimuli (14 per run) and reported the perceived gaps via a button press using a response box Current Designs Trainer 4 Buttons Inline.

## Functional MRI data analysis

### Preprocessing

Field distortion in the functional data was corrected using the top-up tool implemented in FSL (**Smith et al., 2004**). Further data preprocessing was performed in SPM12 (7487). Preprocessing steps included: slice timing correction, motion correction, resampling to 3×3 × 3 mm, coregistration between structural and functional data, normalization to MNI template and 6 mm full-width-at-half-maximum-kernel Gaussian smoothing.

### General linear model

Brain regions responding to the FM stimulus were identified by fitting a general linear model (GLM) to the pre-processed fMRI data for each individual participant. The model was fitted for each individual session and included tree main predictors: FM-stimulus period, detected gap periods (hits) and undetected gap periods (misses), although only the first predictor is relevant for the current analysis. Each predictor was convolved with the canonical hemodynamic response function (HRF). Time and dispersion derivatives of the HRF and a constant term were also included in the model. Additionally, the six realignment parameters were included as nuisance regressors (3 rotation (x, y, z) and 3 translation estimates (x, y, z)). For each individual, BOLD signal during the FM periods was compared to the implicit baseline periods (only fixation without auditory stimulation). Using the individual contrasts, a second level analysis was performed to evaluate the regions responding to the FM-stimulus at the group level. A t-test was performed on the group contrast and individual beta estimates were retained for each participant and hemisphere, as an index of the strength of the activation.

### Region of interest (ROI) definition

Individual T-maps for the contrast FM stimulus > baseline across sessions were thresholded to $p$ $(FWE) < 0.001$, with an extended threshold of 10 voxels and masked to include only temporal regions (including the Heschl's gyrus) according to the automatic anatomical labeling (AAL) atlas. For nine participants, using $p$ $(FWE) < 0.001$ for thresholding t-maps yielded either none or only unilateral activation in auditory regions, therefore uncorrected $p < 0.001$ (8 participants) or $p < 0.05$ (1 participant) were used instead. For each subject, FM-responsive ROI were defined by binarizing the resulting t-maps. The centroid coordinates from the ROIs were estimated for each subject and hemisphere. ROIs for the optimization procedure were defined by creating a 3 mm radius circumference around the centroid coordinates.

## Electric field simulations and montage optimization

### Montage optimization

The electric field simulations and individual montage optimization were implemented in SimNIBS 3.2 (**Saturnino et al., 2019**). First, individual T1 and T2 images were segmented and Finite Element Methods (FEM) head models were implemented using the headreco pipeline. Small segmentation errors were corrected manually in FreeSurfer. The optimization pipeline employed the approach described in **Saturnino et al., 2019** and was performed in two steps. First, a lead field matrix was created per individual using the 10–10 EEG virtual cap provided in SimNIBS and performing electric field simulations based on the default tissue conductivities listed below. The optimization procedure was then performed specifying two 3 mm radius spheres around the center coordinates from the individual ROIs obtained from the fMRI functional localizer as the target ROIs, one per hemisphere. The orientation of the E-field relative to the target ROI was specified to visually resemble an auditory dipole orientation from a former study (**Baltus et al., 2018**). Four electrodes were allowed with a maximum total current of 2 mA. In a first step, the algorithm was requested to optimize for a maximum electric field strength of 0.1 V/m in each target, while minimizing the electric field in the eyes. The latter was done to avoid any retinal stimulation which could induce phosphenes. Since the first optimization step did not perform as desired for all participants, because the requested E-field intensity could not be achieved at both targets, the same procedure was performed a second time but

specifying the desired maximum E-field intensity to be achieved at both targets as the E-field intensity that was achieved in the first step minus 0.01 V/m. Additionally, target E-field intensity was decreased for some participants if visual inspection of the resulting E-field strength maps showed asymmetrical stimulation of the two hemispheres (10 subjects). This was done to assure obtaining a montage that would potentially stimulate both targets with the same intensity. As result, final target E-field strengths for the optimization procedure varied across participants from 0.04 to 0.09 V/m (mean: 0.071, std: 0.018).

### Electric field simulations

For each participant, electric field simulations were performed for three different electrode montages: the individually optimized and two other montages previously used in the literature (*van Bree et al., 2021*; *Baltus et al., 2018*). For each montage, the electric field was simulated using four electrodes, with a maximum current of 0.5 mA, to achieve a current of 1 mA peak-to-peak per channel. The specific locations for placing the electrodes for the individually optimized montage were different across participants. The ring-electrode montage (*van Bree et al., 2021*) included two circular electrodes of 20 mm diameter and 2 mm thickness each surrounded by a ring electrodes of 100 mm diameter and 2 mm thickness with a 75 mm diameter hole. Electrodes were placed over T7 and T8 for all participants. For the standard montage (*Baltus et al., 2018*), electrodes were placed over FC5-TP7/ P7 and FC6-TP8/P8 for all participants. For both the individualized and the standard montages, four circular electrodes were used with 25 mm diameter and 2 mm thickness, two per hemisphere. Electric field simulations were generated for each subject using the default tissue conductivities provided in SimNIBS: $\sigma$skin = 0.465 S/m, $\sigma$skull = 0.010 S/m, $\sigma$CSF = 1.654 S/m, $\sigma$GM = 0.275 S/m, $\sigma$WM = 0.126 S/m, for skin, skull, cerebrospinal-fluid (CSF), gray matter (GM), and white matter (WM), respectively. For each participant and montage, multiple brain maps were obtained including the strength of the electric field and its normal component and data were extracted in MATLAB using customized scripts (*Cabral-Calderin, 2024*) and functions provided by SimNIBS.

## Procedure: tACS sessions

The tACS sessions were conducted in an electrically shielded and acoustically isolated chamber and under normal-illumination conditions. Sound-level thresholds were determined for each participant according to the method of limits. All stimuli were then presented at 55 dB above the individual hearing threshold (55 dB sensation level, SL).

For each session, gap duration was individually adjusted to detection-threshold levels using an adaptive-tracking procedure comprising two interleaved staircases and a weighted up-down technique with custom weights as described in *Cabral-Calderin and Henry, 2022*. This resulted in gap durations ranging from 9 to 24ms across participants and sessions (session 1: mean = 16.805ms, SD = 3.558ms, range = 10–24ms; session 2: mean = 15.205, SD = 3.466, range = 9–22ms).

Before starting the main experiment, participants performed practice trials to make sure they understood the task. For the main experiment, listeners detected gaps embedded in the 20 s long FM stimuli. Listeners were instructed to respond as quickly as possible when they detected a gap via button-press. Participants performed five blocks, each comprising the same set of 32 stimuli (4 per starting phase) but in a different randomized order. Each stimulus included 3–5 gaps for a total of 136 gaps (13–16 per phase bin, mean ~15). The number of phase bins was chosen to balance having a good sampling resolution and a high number of trials per condition without making the task too long.

### Stimulation and data acquisition

Behavioral data were recorded online by MATLAB 2017a (MathWorks) in combination with Psychtoolbox. Sounds were presented at a rate of 44.1 kHz, via an external soundcard (RME Fireface UCX 36-channel, USB 2.0 & FireWire 400 audio interface) using ASIO drivers. Participants listened to the sounds via over-ear headphones (Beyerdynamic DT-770 Pro 80 Ohms, Closed-back Circumaural Dynamic Diffuse field equalization Impedance: 80 Ohm SPL: 96 dB Frequency range: 5–35,000 Hz). Button presses were collected using the computer keyboard.

A battery-driven multichannel Eldith DC-stimulator Plus (NeuroConn GmbH, Ilmenau, Germany) delivered tACS through two pairs of conductive rubber electrodes attached with electrode paste (Weaver and Company, Aurora, CO). Participants received stimulation with one of two different

electrode montages: a standard montage (applied the same way to all participants) and an individualized montage (*Figure 2a*). For the standard montage, electrodes were placed over FC5-TP7/P7 and FC6-TP8/P8 as determined by the International 10–20 EEG system. Note that such montage was previously optimized for targeting the auditory cortex in another study (*Baltus et al., 2018*). For the individualized montage, electrode positions could be different for each participant. Such montage was individually optimized to target bilateral auditory regions of interest, determined in a functional MRI session (see section: *Electric field simulations and montage optimization*; *Figures 1f and 2a*). Each montage comprised four circular electrodes with 25 mm diameter and 2 mm thickness. Each pair of electrodes (one per hemisphere) was connected to a different stimulator channel. A third stimulator channel using the same stimulator waveform but at lower current strength was connected to an EEG amplifier (Brainamp DC amplifiers, Brain Products GmbH) in order to collect the tACS signal for further analysis. The tACS signals from the additional stimulator channel were collected as separate EEG channels using a split ground.

In each session, participants performed 5 blocks of the gap-detection task: 4 with verum tACS and one with sham tACS. Verum tACS was applied at the FM-stimulus frequency (2 Hz) across 4 blocks of 11 min and 20 s each for the duration of each block. The current was fixed to 1 mA (peak-to-peak), and was ramped up and down over the first and final 10 s of each block. Since the FM stimulus always started with a different phase, and the inter-trial interval duration was jittered, the phase lag between the tACS signal and the FM stimulus varied from trial to trial. Sham tACS lasted for one block of 11 min and 20 s, and the serial position of the sham block (out of five total) was randomized across participants and also between sessions within each participant. Sham tACS was ramped on over 10 s to mimic the sensation of the stimulation starting, but was then ramped back down after 10 s of stimulation (ramp down 10 s for a total of 30 s stimulation). The waveform of the stimulation was sinusoidal without DC offset. Impedance was kept below 10 kΩ.

## Data analysis
### Confirming behavioral entrainment in the sham condition
Hits were defined as gaps followed by a button-press response by no earlier than 100 ms and no later than 1.5 s (*Cabral-Calderin and Henry, 2022*), and misses were defined as gaps that were not responded to. First for the sham condition, we calculated gap-detection hit rates as a function of the phase of the FM stimulus in which the gaps were presented, and tested for significance of the predicted FM-induced modulation of gap detection behavior. We fitted a cosine function to hit rates as a function of FM phase for each participant and session. The fitted amplitude parameter (*FM-amplitude*) quantifies the strength of the behavioral modulation by 2 Hz FM phase. The optimal FM phase (*FM-phase*) is the phase that yields peak predicted gap-detection hit rate. The intercept parameter corresponds to mean hit rate across FM phases, and was not analyzed further here. Significance of the behavioral modulation (*FM-amplitude*) was tested using a permutation approach, whereby 1000 surrogate datasets were created for each participant and session by shuffling the single-gap accuracy values (0,1) with respect to their FM-stimulus phase bin labels. Cosine functions were also fitted to the surrogate datasets. Gap detection was considered to be significantly modulated for each participant if the individual FM-amp value was higher than the 95th percentile of the distribution of FM-amp values from the surrogate data, corresponding to p<0.05.

### Evaluating tACS effects
In order to determine the tACS phase timecourse for each trial in the verum stimulation blocks, the tACS signal collected in the EEG was band-pass filtered between 1 and 10 Hz, submitted to a Hilbert transform, and the complex output was converted to phase angles (4-quadrant inverse tangent, *atan2*). For each gap, the phase lag between the 2 Hz FM stimulus and the 2 Hz tACS signal was estimated as the circular distance between their instantaneous phases at the 2 Hz FM stimulus onset. Individual gaps were then sorted into 6 FM–tACS phase-lags bins centered on 0, $\pi/3$, $2\pi/3$, $\pi$, $-2\pi/3$, and $-\pi/3$ (see *Figure 3a*). Hit rates were calculated separately for each of the nine FM-phase bins and the six FM–tACS lags (plus the sham condition, as described above). Cosine fits were used to estimate *FM-amplitude* and *FM-phase* for each participant, session, and FM–tACS lag.

The effect of FM–tACS lag on FM-phase was evaluated by investigating whether *FM-phase* were clustered around the same value for all tACS conditions using the V-test as implemented in the circular

statistic toolbox (*Berens, 2009*). To estimate the effect of FM–tACS lag on *FM-amplitude*, a second-level cosine function was fitted to the *FM-amplitude* values as a function of FM–tACS lag. From this second-level fit, the optimal tACS phase lag (*tACS-phase*) was defined as the FM–tACS lag yielding the highest predicted *FM-amplitude* value.

## Evaluating tACS effects at the group level

*TACS-phase* values were highly variable across individuals. Thus, in order to fairly evaluate FM–tACS effects on *FM-amplitude* at the group level, FM-amplitude values were circularly realigned so that optimal tACS-phase was zero-phase for each participant and session. To control for possible analytical biases induced by the alignment procedure, *FM-amplitude* at the individual optimal tACS lag and *FM-amplitude* at the opposite phase were excluded from further analysis (*Riecke et al., 2015*). We statistically tested mean *FM-amplitude* values for positive against negative realigned phase values: *FM-amplitude* values for positive FM–tACS lags (tACS(+)) were *FM-amplitude* averaged across the two tACS lags adjacent to the aligned zero-lag condition, and *FM-amplitude* values for negative FM–tACS lags (tACS(-)) were *FM-amplitude* averaged across the two tACS lags adjacent to the tACS lag opposite the realigned zero-lag condition. Group tACS effects were then assessed using a mixed ANOVA with the repeated measures tACS lag condition (3 levels: sham, tACS(+), and tACS(-)) and session (2 levels: S1 and S2) and the tACS montage as between factor. In addition, we modelled all two-way and three-way interactions.

To further control that the observed tACS effects were not an artifact of the analysis procedure, the difference between the tACS conditions (sham, tACS(+), and tACS(-)) were normalized using a permutation approach. For each participant and session, 1000 surrogate datasets were created by permuting the tACS lag designation across trials. The same binning procedure, realignment, and cosine fits were applied to each surrogate dataset as for the original data (see above). FM-amplitude at sham, tACS(+) and tACS(-) were averaged across sessions since the original analysis did not show a main effect of session. Difference between tACS conditions were estimated for the original and surrogate datasets and the resulting values from the original data were z-scored using the mean and standard deviation from the surrogate distributions. One-sample t-tests were conducted to test the statistical significance of the z-scores. p-Values were corrected for multiple comparisons using the *Holm-Bonferroni* method.

## Evaluating the reliability of tACS effects

Reliability of the strength of tACS effects was investigated by comparing *tACS-amplitude* values between sessions as well as via Pearson's correlation coefficient. Reliability of *tACS-phase* was assessed by computing the circular distance between *tACS-phase* across sessions and testing its distribution against zero. In addition, circular-to-circular correlation of tACS-phase across sessions was employed.

## Predicting tACS effects with linear models

The difference between FM-amplitude at the positive half of the tACS cycle (tACS(+)) and the FM-amplitude at the negative half of the tACS cycle (tACS(-)) was first calculated for each subject and session and then averaged across sessions. This average difference was taken as the overall tACS effect (*tACS-amplitude*) for each participant. We aimed to test the influence of the electric field characteristics and baseline performance (i.e. *FM-amplitude* at sham, auditory-driven BOLD signal increase) on tACS effects. To do this, we fit several linear models using the MATLAB function '*fitlm*'. Each model evaluated whether inter-individual variability in tACS effects could be predicted as a function of seven parameters extracted from the simulated electric field: (1) mean Euclidian distance between the center coordinates of the functional ROIs (one per hemisphere) and the center coordinates of the peak of the electric field in each hemisphere (where the peak is for the 99-percentile-thresholded E-field strength across all voxels for the given hemisphere), (2) spatial Pearson correlation coefficient between the individual BOLD signal t-map and the electric field map, (3) peak E-field strength for the whole brain (based on the 95-percentile-thresholded E-field map), (4) mean peak E-field strength in the functional ROIs (based on the 95-percentile-thresholded E-field from a 7 mm sphere around the center coordinates of the functional ROIs), (5) peak normal component of the E-field for the whole brain (95-percentile-thresholded), (6) peak normal component of the E-field in the functional ROIs (defined as in 4) and (7) E-field focality (cortical area in mm² with E-field strength higher than the

50th percentile). Additional regressors in the model included baseline predictors: (8) amplitude of the FM-induced behavioral modulation at sham, averaged over sessions, (9) mean beta estimates from the BOLD signal map; and the interaction between the terms. All predictors were transformed to z-scores prior to model fitting. Before z-score transformation, the field focality and the *Dist2Peak* predictors were inverted so higher values mean higher focality and shorter distance. Thirty-three different models were fitted, each including a different combination of regressors, and the best model was selected using the Akaike's information criterion corrected for small samples (AICc, **Supplementary file 1e**) and the Likelihood Ratio Test (LRT).

Additional linear models were estimated using the same procedure as describe above but to evaluate the effect of regressors (1) gender, (2) age (3) number of days between sessions, ΔDays, (4) inter-session time on the day difference in minutes, ΔMinutes, (5) inter-session difference in gap size threshold and (6) montage on the inter-session reliability of tACS effects: inter-session difference in *tACS-amplitude* (session 1- session 2) and absolute circular distance in tACS-phase. For each dependent variable, multiple models were tested (**Supplementary file 1b-c**). Significance and model selection were performed as above.

## Evaluating the effect of the electrode montage on inter-individual variability

A variance test was performed on the *tACS-amplitude* values averaged across sessions and grouped by tACS montage to test for the effect of montage on inter-subject variability in the overall tACS effect. Additional variance tests were performed on the *FM-amplitude* values for tACS(+) and tACS(-) separated by session to check for montage-specific differences in inter-individual variability specific to a given tACS condition and session. Resultant p-values were corrected for 4 comparisons using the Bonferroni method. For descriptive information on inter-individual variability for each group and condition, the coefficient of variation (CV) and the interquartile ratio (iqr) were calculated.

### Other statistical Analyses

Unless otherwise specified, correlation analyses between linear variables were done using Pearson correlation coefficient. In case of circular data, circular–circular or circular–linear correlations were computed using the respective functions in the circular statistics toolbox for MATLAB (**Berens, 2009**). Differences between tACS montages in terms of the simulated electric fields were investigated using rmANOVAs. Post-hoc analyses were conducted using standard paired t-tests. Significant p-values were corrected using *Bonferroni* method.

Behavioral data generated and analyzed during this study are publicly shared in the OSF platform: https://osf.io/k5fwp/. Custom MATLAB scripts are made available via GitHub ( https://github.com/ycabralcalderin/Cabral-Calderin_et_al_eLife_2023/releases/tag/MATLAB_code_tACS_FMaudio_BehEntrainment copy archived at *Cabral-Calderin, 2024*).

## Acknowledgements

We thank Hanna Kadel, Dominik Thiele, and Cornelius Abel for technical support. We thank the MRI core Unit of the Brain Imaging Center in Frankfurt for technical support setting up the MRI sequences. This work was supported by a European Research Council Starting Grant (BRAINSYNC) and a Max Planck Research Group granted to MJH. AT was supported by the Lundbeck foundation (grants R244-2017-196 and R313-2019-622).

## Additional information

### Funding

| Funder | Grant reference number | Author |
|---|---|---|
| Lundbeck Foundation | R244-2017-196/R313-2019-622 | Axel Thielscher |
| European Research Council | BRAINSYNC | Molly J Henry |

| Funder | Grant reference number | Author |
|---|---|---|

The funders had no role in study design, data collection and interpretation, or the decision to submit the work for publication. Open access funding provided by Max Planck Society.

## Author contributions

Yuranny Cabral-Calderin, Conceptualization, Resources, Data curation, Software, Formal analysis, Supervision, Validation, Investigation, Visualization, Methodology, Writing – original draft, Project administration, Writing – review and editing; Daniela van Hinsberg, Investigation, Project administration, Writing – review and editing; Axel Thielscher, Methodology, Writing – review and editing; Molly J Henry, Conceptualization, Resources, Software, Supervision, Funding acquisition, Validation, Methodology, Writing – review and editing

## Author ORCIDs

Yuranny Cabral-Calderin ⬥ https://orcid.org/0000-0002-5497-9360
Molly J Henry ⬥ https://orcid.org/0000-0002-2284-8884

## Ethics

Written informed consent was obtained from all participants. The procedure was approved by the Ethics Council of the Max Planck Society (registration No. 2019_27) and in accordance with the declaration of Helsinki.

Reviewer #1 (Public Review): https://doi.org/10.7554/eLife.87820.3.sa1
Reviewer #2 (Public Review): https://doi.org/10.7554/eLife.87820.3.sa2
Author Response https://doi.org/10.7554/eLife.87820.3.sa3

# Additional files

## Supplementary files

• Supplementary file 1. Tables. (a). MNI center coordinates for target functional regions of interest for each subject. (b) Statistics for the mixed effects logistic regression models predicting single trial gap detection performance. Models are organized from smallest to highest AIC. Δ AIC relative to winning model. The winning model is also highlighted in bold. BIC: Bayesian information criterion. (c) General linear models predicting inter-session difference on *tACS-amplitude*. Models are organized from smallest to highest AICc. * Δ AICc relative to winning model. The winning model is also highlighted in bold. (d) General linear models predicting inter-session absolute circular distance on *tACS-phase*. Models are organized from smallest to highest AICc. * Δ AICc relative to winning model. The winning model is also highlighted in bold. (e) General linear models predicting tACS effects. Models are organized from smallest to highest AICc. * Δ AICc relative to winning model. The winning model is also highlighted in bold.

• MDAR checklist

## Data availability

Behavioral data generated and analyzed during this study are publicly shared in the OSF platform: https://osf.io/k5fwp/.

The following dataset was generated:

| Author(s) | Year | Dataset title | Dataset URL | Database and Identifier |
|---|---|---|---|---|
| Cabral-Calderin Y | 2022 | dataGroupallGaps.mat | https://osf.io/k5fwp/ | Open Science Framework, 10.17605/OSF.IO/K5FWP |

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
