## [Editor Report · eLife assessment]

This detailed and well powered manuscript explores auditory perception of modulated noise in the presence of transcranial alternating-current stimulation (tACS) and shows **valuable** results suggesting that there are subject-specific effects when the phase of 2-Hz tACS varies relative to the phase of the noise modulation. The strength of the evidence is mixed. There is **convincing** evidence that tACS alters perception significantly in individuals; however, the effects are inconsistent across subjects and even across sessions, frustrating attempts to draw conclusions about the underlying mechanisms of the idiosyncratic effects. Despite these limitations, the paper will be of great interest to researchers interested in determining when and how tACS influences neural processes, especially those interested in neural entrainment and its relationship to perception.

---

## [Referee Report · Reviewer #1 (Public Review)]

This paper studies the effects of tACS on detection of silence gaps in an FM modulated noise stimulus. Both FM modulation of the sound and the tACS are at 2Hz, and the phase of the two is varied to determine possible interactions between the auditory and electric stimulation. Additionally, two different electrode montages are used to determine if variation in electric field distribution across the brain may be related to the effects of tACS on behavioral performance in individual subjects.

Major strengths and weaknesses of the methods and results.

The study appears to be well powered to detect modulation of behavioral performance with N=42 subjects. There is a clear and reproducible modulation of behavioral effects with the phase of the FM sound modulation. The study was also well designed and executed in terms of fMRI, current flow modeling, montage optimization targeting, and behavioral analysis. A particular merit of this study is to have repeated the sessions for most subjects in order to test repeat-reliability, which is so often missing in human experiments. The results and methods are generally well described and well conceived. The portion of the analysis related to behavior alone is excellent. The analysis of the tACS results are also generally well described, candidly highlighting how variable results are across subjects and sessions. The figures are all of high quality and clear. One weakness of the experimental design is that no effort was made to control for sensation effects. tACS at 2Hz causes prominent skin sensations which could have interacted with auditory perception and thus, detection performance.

The central claim is that tACS modulates behavioral detection performance across the 0.5s cycle of stimulation. Statistical analysis with randomize relative phase (between audio and tACS) show that detection performance is modulated by tACS. Neither the relative phase or the strength of this effect reproduces across subjects or sessions, which makes the interpretation of these results difficult. These result could be of interest to investigators in the field of tACS.

The claim that the variation in the strength of the effect can be explained by variation of electric fields is not compelling.

The following are more detailed comments to specific sections of the paper, including details on the concerns with the statistical analysis of the tACS effects.

The introduction is well balanced, discussing the promise and limitations of previous results with tACS. The objectives are well defined.

The analysis surrounding behavioral performance and its dependence on phase of the FM modulation (Figure 3) is masterfully executed and explained. It appears that it reproduces previous studies and points to a very robust behavioral task that may be of use in other studies.

The definition of tACS(+) vs tACS(-) phase is adjusted to each subject/session, which seems unconventional. For argument sake, let's assume the curves in Fig. 3E are random fluctuations. Then aligning them to best-fitting cosine will trivially generate a FM-amplitude fluctuation with cosine shape as shown in Fig. 4a. Selecting the positive and negative phase of that will trivially be larger and smaller than sham, respectively, as shown in Fig 4b.

"Data from the optimal tACS lag and its opposite lag (corresponding trough) were excluded to avoid any artificial bias in estimating tACS effects induced by the alignment procedure (33)." The delay was found by fitting a cosine, so removing just the peaks of that cosine does little to avoid this problem.

To demonstrate that this is not a trivial result of the definition, the analysis compares this to the same analysis but with a randomize alignment to the two stimuli (audio and tACS) in Figure 4d. Assuming this shuffle was done correctly, this shows that the modulation observed in 4b is not just a result of the analysis procedure.

The authors are to be commended for analyzing the robustness of their observation across subjects and across sessions in Fig. 5. The lack of consistency in the optimal time delay between the two stimuli is hard to reconcile with the common theory that tACS entrains brain function.

"To better understand what factors might be influencing inter-session variability in tACS effects, we estimated multiple linear models ..." "Inter-individual variability in the simulated E-field predicts tACS effects" Authors here are attempting to predict a property of the subjects that was just shown to not be a reliable property of the subject. Authors are picking 9 possible features for this, testing 33 possible models with N=34 data points. With these circumstances it is not hard to find something that correlates by chance. And some of the models tested had interaction terms, possibly further increasing the number of comparisons. In the absence of multiple comparison correction, what is happening here is that multiple models are fit to the data, and a statistical test is performed for the best model on the same (training) data. The corresponding claim that variations are explained by variations in electric field is not persuasive.

"Can we reduce inter-individual variability in tACS effects ..." This section seems even more speculative and with mixed results.

Given the concerns with the statistical analysis above, there are concerns about the following statements in the summary of the Discussion:

("4) individual variability in tACS effect size was partially explained by two interactions: between the normal component of the E-field and the field focality, and between the normal component of the E-field and the distance between the peak of the electric field and the functional target ROIs."

The complexity of this statement alone may be a good indication that this could be the result of false discovery due to multiple comparisons.

For the same reason as stated above, the following statements in the Abstract do not appear to have adequate support in the data:

"Inter-individual variability of tACS effects was best explained by the strength of the inward electric field, depending on the field focality and proximity to the target brain region. Although additional evidence is necessary, our results

42 also provided suggestive insights that spatially optimizing the electrode montage could be a promising tool to reduce inter-individual variability of tACS effects."

---

## [Referee Report · Reviewer #2 (Public Review)]

I thank the authors for considering my comments and think the manuscript has been significantly improved with revision. However while I considered that the analysis employed for predicting tACS effects with linear models was convincing, I am still concerned by a multiple comparison issue for this analysis. An alternative option would be to report the results of a Partial Least Squares (PLS) analysis, with the stimulation properties as predictor variables and tACS effects as response variables. The authors could use PLS instead of multiple linear regression models to take into account the multicollinearity in the predictor variables, and also this can be done with only one PLS model. They could then extract the fitted responses values and estimate if the model can significantly fit the tACS effects.

Then, to determine which variables contribute more to the prediction, they can calculate the variable importance in projection (VIP) scores for the PLS regression model.

An alternative option for the authors would be to temper their conclusions regarding how well field modeling/montage explains the variance observed across subjects.

---

## [Author Response]

The following is the authors’ response to the original reviews.

Thank you for organizing the reviews for our manuscript: Behavioral entrainment to rhythmic auditory stimulation can be modulated by tACS depending on the electrical stimulation field properties,” and for the positive eLife assessment. We also thank the reviewers for their constructive comments. We have addressed every comment, which has helped to improve the transparency and readability of the manuscript. The main changes to the manuscript are summarized as follows:

1. Surrogate distributions were created for each participant and session to estimate the effect of tACS-phase lag on behavioral entrainment to the sound that could have occurred by chance or because of our analysis method (R1). The actual tACS-amplitude effects were normalized relative to the surrogate distribution, and statistical analysis was performed on the normalized (z-score) values. This analysis did not change our main outcome: that tACS modulates behavioral entrainment to the sound depending on the phase lag between the auditory and the electrical signals. This analysis has now been incorporated into the Results section and in Fig. 3c-d.

2. Two additional supplemental figures were created to include the single-participant data related to Fig. 3b and 3e (R2).

3. Additional editing of the manuscript has been performed to improve the readability.

Below, you will find a point-by-point response to the reviewers’ comments.

**Reviewer #1 (Public Review):**

We are grateful for the reviewer’s positive assessment of the potential impact of our study. The reviewer’s primary concerns were (1) the tACS lag effects reported in the manuscript might be noise because of the realignment procedure, and (2) no multiple comparisons correction was conducted in the model comparison procedure.

In response to point (1), we have reanalyzed the data in exactly the manner prescribed by the reviewer. Our effects remain, and the new control analysis strengthens the manuscript. (2) In the context of model comparison, the model selection procedure was not based on evaluating the statistical significance of any model or predictor. Instead, the single model that best fit the data was selected as the model with the lowest Akaike’s information criterion (AIC), and its superiority relative to the second-best model was corroborated using the likelihood ratio test. Only the best model was evaluated for significance and analyzed in terms of its predictors and interactions. This model is an omnibus test and does not require multiple comparison correction unless there are posthoc decompositions. For similar approaches, see (Kasten et al., 2019).

Below, we have responded to each comment specifically or referred to this general comment.

Summary of what the authors were trying to achieve.This paper studies the possible effects of tACS on the detection of silence gaps in an FM-modulated noise stimulus. Both FM modulation of the sound and the tACS are at 2Hz, and the phase of the two is varied to determine possible interactions between the auditory and electric stimulation. Additionally, two different electrode montages are used to determine if variation in electric field distribution across the brain may be related to the effects of tACS on behavioral performance in individual subjects.Major strengths and weaknesses of the methods and results.The study appears to be well-powered to detect modulation of behavioral performance with N=42 subjects. There is a clear and reproducible modulation of behavioral effects with the phase of the FM sound modulation. The study was also well designed, combining fMRI, current flow modeling, montage optimization targeting, and behavioral analysis. A particular merit of this study is to have repeated the sessions for most subjects in order to test repeat-reliability, which is so often missing in human experiments. The results and methods are generally well-described and well-conceived. The portion of the analysis related to behavior alone is excellent. The analysis of the tACS results is also generally well described, candidly highlighting how variable results are across subjects and sessions. The figures are all of high quality and clear. One weakness of the experimental design is that no effort was made to control for sensation effects. tACS at 2Hz causes prominent skin sensations which could have interacted with auditory perception and thus, detection performance.

The reviewer is right that we did not control for the sensation effects in our paradigm. We asked the participants to rate the strength of the perceived stimulation after each run. However, this information was used only to assess the safety and tolerability of the stimulation protocol. Nevertheless, we did not consider controlling for skin sensations necessary given the within-participant nature of our design (all participants experienced all six tACS–audio phase lag conditions, which were identical in their potential to cause physical sensations; the only difference between conditions was related to the timing of the auditory stimulus). That is, while the reviewer is right that 2-Hz tACS can indeed induce skin sensation under the electrodes, in this study, we report the effects that depend on the tACS-phase lag relative to the FM-stimulus. Note that the starting phase of the FM-stimulus was randomized across trials within each block (all six tACS audio lags were presented in each block of stimulation). We have no reason to expect the skin sensation to change with the tACS-audio lag from trial to trial, and therefore do not consider this to be a confound in our design. We have added some sentences with this information to the Discussion section:

Pages 16-17, lines 497-504: “Note that we did not control for the skin sensation induced by 2-Hz tACS in this experiment. Participants rated the strength of the perceived stimulation after each run. However, this information was used only to assess the safety and tolerability of the stimulation protocol. It is in principle possible that skin sensation would depend on tACS phase itself. However, in this study, we report effects that depend on the relationship between tACS-phase and FM-stimulus phase, which changed from trial to trial as the starting phase of the FM-stimulus was randomized across trials. We have no reason to expect the skin sensation to change with the tACS-audio lag and therefore do not consider this to be a confound in our data.”

Appraisal of whether the authors achieved their aims, and whether the results support their conclusions.Unfortunately, the main effects described for tACS are encumbered by a lack of clarity in the analysis. It does appear that the tACS effects reported here could be an artifact of the analysis approach. Without further clarification, the main findings on the tACS effects may not be supported by the data.Likely impact of the work on the field, and the utility of the methods and data to the community.The central claim is that tACS modulates behavioral detection performance across the 0.5s cycle of stimulation. However, neither the phase nor the strength of this effect reproduces across subjects or sessions. Some of these individual variations may be explainable by individual current distribution. If these results hold, they could be of interest to investigators in the tACS field.The additional context you think would help readers interpret or understand the significance of the work.The following are more detailed comments on specific sections of the paper, including details on the concerns with the statistical analysis of the tACS effects.The introduction is well-balanced, discussing the promise and limitations of previous results with tACS. The objectives are well-defined.The analysis surrounding behavioral performance and its dependence on the phase of the FM modulation (Figure 3) is masterfully executed and explained. It appears that it reproduces previous studies and points to a very robust behavioral task that may be of use in other studies.

Again, we would like to thank the reviewer for the positive assessment of the potential impact of our work and for the thoughtful comments regarding the methodology. For readability in our responses, we have numbered the comments below.

1. There is a definition of tACS(+) vs tACS(-) based on the relative phase of tACS that may be problematic for the subsequent analysis of Figures 4 and 5. It seems that phase 0 is adjusted to each subject/session. For argument's sake, let's assume the curves in Fig. 3E are random fluctuations. Then aligning them to best-fitting cosine will trivially generate a FM-amplitude fluctuation with cosine shape as shown in Fig. 4a. Selecting the positive and negative phase of that will trivially be larger and smaller than a sham, respectively, as shown in Fig 4b. If this is correct, and the authors would like to keep this way of showing results, then one would need to demonstrate that this difference is larger than expected by chance. Perhaps one could randomize the 6 phase bins in each subject/session and execute the same process (fit a cosine to curves 3e, realign as in 4a, and summarize as in 4b). That will give a distribution under the Null, which may be used to determine if the contrast currently shown in 4b is indeed statistically significant.

We agree with the reviewer’s concerns regarding the possible bias induced by the realignment procedure used to estimate tACS effects. Certainly, when adjusting phase 0 to each participant/session’s best tACS phase (peak in the fitting cosine), selecting the positive phase of the realigned data will be trivially larger than sham (Fig. 4a). This is why the realigned zero-phase and opposite phase (trough) bins were excluded from the analysis in Fig. 4b. Therefore, tACS(+) vs. tACS(-) do not represent behavioral entrainment at the peak positive and negative tACS lags, as both bins were already removed from the analysis. tACS(+) and tACS(-) are the averages of two adjacent bins from the positive and negative tACS lags, respectively (Zoefel et al., 2019). Such an analysis relies on the idea that if the effect of tACS is sinusoidal, presenting the auditory stimulus at the positive half cycle should be different than when the auditory stimulus lags the electrical signal by the other half. If the effect of tACS was just random noise fluctuations, there is no reason to assume that such fluctuations would be sinusoidal; therefore, any bias in estimating the effect of tACS should be removed when excluding the peak to which the individual data were realigned. Similar analytical procedures have been used previously in the literature (Riecke et al., 2015; Riecke et al., 2018). We have modified the colors in Fig. 4a and 4c (former 4b) and added a new panel to the figure (new 4b) to make the realignment procedure, including the exclusion of the realigned peak and trough data, more visually obvious.

Moreover, we very much like the reviewer’s suggestion to normalize the magnitude of the tACS effect using a permutation strategy. We performed additional analyses to normalize our tACS effect in Fig. 4c by the probability of obtaining the effect by chance. For each subject and session, tACS-phase lags were randomized across trials for a total of 1000 iterations. For each iteration, the gaps were binned by the FM-stimulus phase and tACS-lag. For each tACS-lag, the amplitude of behavioral entrainment to the FM-stimulus was estimated (FM-amplitude), as shown in Fig. 3. Similar to the original data, a second cosine fit was estimated for the FM-amplitude by tACS-lag. Optimal tACS-phase was estimated from the cosine fit and FM-amplitude values were realigned. Again, the realigned phase 0 and trough were removed from the analysis, and their adjacent bins were averaged to obtain the FM-amplitude at tACS(+) and tACS(−), as shown in Fig. 4c. We then computed the difference between (1) tACS(+) and sham, (2) tACS(-) and sham, and (3) tACS(+) and tACS (-), for the original data and the permuted datasets. This procedure was performed for each participant and session to estimate the size of the tACS effect for the original and surrogate data. The original tACS effects were transformed to z-scores using surrogate distributions, providing us with an estimate of the size of the real effect relative to chance. We then computed one-sample t-tests to compare whether the effects of tACS were statistically significant. In fact, this analysis showed that the tACS effects were still statistically significant. This analysis has been added to the Results and Methods sections and is included in Figure 4d.

Page 10, lines 282-297: “In order to further investigate whether the observed tACS effect was significantly larger than chance and not an artifact of our analysis procedure (33), we created 1000 surrogate datasets per participant and session by permuting the tACS lag designation across trials. The same binning procedure, realignment, and cosine fits were applied to each surrogate dataset as for the original data. This yielded a surrogate distribution of tACS(+) and tACS(-) values for each participant and session. These values were averaged across sessions since the original analysis did not show a main effect of session. We then computed the difference between tACS(+) and sham, tACS(-) and sham, and tACS(+) and tACS(-), separately for the original and surrogate datasets. The obtained difference for the original data where then z-scored using the mean and standard deviation of the surrogate distribution. Note that in this case we used data of all 42 participants who had at least one valid session (37 participants with both sessions). Three one-sample t-tests were conducted to investigate whether the size of the tACS effect obtained in the original data was significantly larger than that obtained by chance (Fig. 4d). This analysis showed that all z-scores were significantly higher than zero (all t(41) > 2.36, p < 0.05, all p-values corrected for multiple comparisons using the Holm-Bonferroni method).”

Page 31, lines 962-972: “To further control that the observed tACS effects were not an artifact of the analysis procedure, the difference between the tACS conditions (sham, tACS(+), and tACS(-)) were normalized using a permutation approach. For each participant and session, 1000 surrogate datasets were created by permuting the tACS lag designation across trials. The same binning procedure, realignment, and cosine fits were applied to each surrogate dataset as for the original data (see above). FM-amplitude at sham, tACS(+) and tACS(-) were averaged across sessions since the original analysis did not show a main effect of session. Difference between tACS conditions were estimated for the original and surrogate datasets and the resulting values from the original data were z-scored using the mean and standard deviation from the surrogate distributions. One-sample t-tests were conducted to test the statistical significance of the z-scores. P-values were corrected for multiple comparisons using the Holm-Bonferroni method.”

1. Results of Fig 5a and 5b seem consistent with the concern raised above about the results of Fig. 4. It appears we are looking at an artifact of the realignment procedure, on otherwise random noise. In fact, the drop in "tACS-amplitude" in Fig. 5c is entirely consistent with a random noise effect.

Please see our response to the comment above.

1. To better understand what factors might be influencing inter-session variability in tACS effects, we estimated multiple linear models ..." this post hoc analysis does not seem to have been corrected for multiple comparisons of these "multiple linear models". It is not clear how many different things were tried. The fact that one of them has a p-value of 0.007 for some factors with amplitude-difference, but these factors did not play a role in the amplitude-phase, suggests again that we are not looking at a lawful behavior in these data.

We suspect that the reviewer did not have access to the supplemental materials where all tables (relevant here is Table S3) are provided. This post hoc analysis was performed as an exploratory analysis to better understand the factors that could influence the inter-session variability of tACS effects. In Table S3, we provide the formula for each of the seven models tested, including their Akaike information criteria corrected for small samples (AICc), R2, F, and p-values. As described in the methods section, the winning model was selected as the model with the smallest AICc. A similar procedure has been previously used in the literature (Kasten et al., 2019). Moreover, to ensure that our winning model was better at explaining the data than the second-best unrestricted model, we used the likelihood ratio test. After choosing the winning model and before reporting the significance of the predictors, we examined the significance of the model in and of itself, taking into account its R2 as well as F- and p-values relative to a constant model. Thus, only one model is being evaluated in terms of statistical significance. Therefore, to our understanding, there are no multiple comparisons to correct for. We added the information regarding the selection procedure, hoping this will make the analysis clearer.

See page 12, lines 354-360: “This model was selected because it had the smallest Akaike’s information criterion (corrected for small samples), AICc. Moreover, the likelihood ratio test showed no evidence for choosing the more complex unrestricted model (stat = 2.411, p = 0.121). Following the same selection criteria, the winning model predicting inter-session variability in tACS-phase, included only the factor gender (Table S4). However, this model was not significant in and of itself when compared to a constant model (F-statistic vs. constant model: 3.05, p = 0.09, R2 = 0.082).”

1. "So far, our results demonstrate that FM-stimulus driven behavioral modulation of gap detection (FM-amplitude) was significantly affected by the phase lag between the FM-stimulus and the tACS signal (Audio-tACS lag) ..." There appears to be nothing in the preceding section (Figures 4 and 5) to show that the modulation seen in 3e is not just noise. Maybe something can be said about 3b on an individual subject/session basis that makes these results statistically significant on their own. Maybe these modulations are strong and statistically significant, but just not reproducible across subjects and sessions?

Please see our response to the first comment regarding the validity of our analysis for proving the significant effect of tACS lag on modulating behavioral entrainment to the FM-stimulus (FM-amplitude), and the new control analysis. After performing the permutation tests, to make sure the reported effects are not noise, our statistical analysis still shows that tACS-lag does significantly modulate behavioral entrainment to the sound (FM-amplitude). Thus, the reviewer is right to say “these modulations are strong and statistically significant, just not reproducible across subjects and sessions”. In this regard, we consider our evaluation of session-to-session reliability of tACS effects is of high relevance for the field, as this is often overlooked in the literature.

1. "Inter-individual variability in the simulated E-field predicts tACS effects" Authors here are attempting to predict a property of the subjects that was just shown to not be a reliable property of the subject. Authors are picking 9 possible features for this, testing 33 possible models with N=34 data points. With these circumstances, it is not hard to find something that correlates by chance. And some of the models tested had interaction terms, possibly further increasing the number of comparisons. The results reported in this section do not seem to be robust, unless all this was corrected for multiple comparisons, and it was not made clear?

We thank the reviewer very much for this comment. While the reviewer is right that in these models, we are trying to predict an individual property (tACS-amplitude) that was not test–retest reliable across sessions, we still consider this to be a valid analysis. Here, we take the tACS-amplitude averaged across sessions, trying to predict the probability of a participant to be significantly modulated by tACS, in general, regardless of day-to-day variability. Regarding the number of multiple regression models, how we chose the winning model and the appropriateness/need of multiple-comparisons correction in this case, please see our explanation under “Reviewer 1 (Public review)” and our response to comment 3.

1. "Can we reduce inter-individual variability in tACS effects ..." This section seems even more speculative and with mixed results.

We agree with the reviewer that this section is a bit speculative. We are trying to plant some seeds for future research can help move the field forward in the quest for better stimulation protocols. We have added a sentence at the end of the section to explicitly say that more evidence is needed in this regard.

Page 14, lines 428-429: “At this stage, more evidence is needed to prove the superiority of individually optimized tACS montages for reducing inter-individual variability in tACS effects.”

Given the concerns with the statistical analysis above, there are concerns about the following statements in the summary of the Discussion:

1. ("2) does modulate the amplitude of the FM-stimulus induced behavioral modulation (FM-amplitude)"This seems to be based on Figure 4, which leaves one with significant concerns.

Please see response to comment 1. We hope the reviewer is satisfied with our additional analysis to make sure the effect of tACS here reported is not noise.

1. ("4) individual variability in tACS effect size was partially explained by two interactions: between the normal component of the E-field and the field focality, and between the normal component of the E-field and the distance between the peak of the electric field and the functional target ROIs."The complexity of this statement alone may be a good indication that this could be the result of false discovery due to multiple comparisons.

We respectfully disagree with the reviewer’s opinion that this is a complex statement. We think that these interaction effects are very intuitive as we explain in the results and discussion sections. These significant interactions show that for tACS to be effective, it matters that current gets to the right place and not to irrelevant brain regions. We believe this finding is of great importance for the field, since most studies on the topic still focus mostly on predicting tACS effects from the absolute field strength and neglect other properties of the electric field.

For the same reasons as stated above, the following statements in the Abstract do not appear to have adequate support in the data:"We observed that tACS modulated the strength of behavioral entrainment to the FM sound in a phase-lag specific manner. ... Inter-individual variability of tACS effects was best explained by the strength of the inward electric field, depending on the field focality and proximity to the target brain region. Spatially optimizing the electrode montage reduced inter-individual variability compared to a standard montage group."

Please see response to all previous comments

In particular, the evidence in support of the last sentence is unclear. The only finding that seems related is that "the variance test was significant only for tACS(-) in session 2". This is a very narrow result to be able to make such a general statement in the Abstract. But perhaps this can be made clearer.

We changed this sentence in the abstract to:

Page 2, lines 41-43: “Although additional evidence is necessary, our results also provided suggestive insights that spatially optimizing the electrode montage could be a promising tool to reduce inter-individual variability of tACS effects.”

**Reviewer #3 (Public Review):**
In "Behavioral entrainment to rhythmic auditory stimulation can be modulated by tACS depending on the electrical stimulation field properties" Cabral-Calderin and collaborators aimed to document (1) the possible advantages of personalized tACS montage over standard montage on modulating behavior; (2) the inter-individual and inter-session reliability of tACS effects on behavioral entrainment and, (3) the importance of the induced electric field properties on the inter-individual variability of tACS.To do so, in two different sessions, they investigated how the detection of silent gaps occurring at random phases of a 2Hz- amplitude modulated sound could be enhanced with 2Hz tACS, delivered at different phase lags. In addition, they evaluated the advantage of using spatially optimized tACS montages (information-based procedure - using anatomy and functional MRI to define the target ROI and simulation to compare to a standard montage applied to all participants) on behavioral entrainment. They first show that the optimized and the standard montages have similar spatial overlap to the target ROI. While the optimized montage induced a more focal field compared to the standard montage, the latter induced the strongest electric field. Second, they show that tACS does not modify the optimal phase for gap detection (phase of the frequency-modulated sound) but modulates the strength of behavioral entrainment to the frequency-modulated sound in a phase-lag specific manner. However, and surprisingly, they report that the optimal tACS lag, and the magnitude of the phasic tACS effect were highly variable across sessions. Finally, they report that the inter-individual variability of tACS effects can be explained by the strength of the inward electric field as a function of the field focality and on how well it reached the target ROI.The article is interesting and well-written, and the methods and approaches are state-of-the-art.Strengths:The information-based approach used by the authors is very strong, notably with the definition of subject-specific targets using a fMRI localizer and the simulation of electric field strength using 3 different tACS montages (only 2 montages used for the behavioral experiment).The inter-session and inter-individual variability are well documented and discussed. This article will probably guide future studies in the field.Weaknesses:The addition of simultaneous EEG recording would have been beneficial to understand the relationship between tACS entrainment and the entrainment to rhythmic auditory stimulation.

We are grateful for the Reviewer’s positive assessment of our work and for the reviewer’s recommendations. We agree with the reviewer that adding simultaneous EEG or MEG to our design would have been beneficial to understand tACS effects. However, as the reviewer might be familiar with, such combination also possesses additional challenges due to the strong artifacts induced by tACS in the EEG signals, which is at the frequency of interest and several orders of magnitude higher than the signal of interest. Unfortunately, the adequate setup for simultaneous tACS-EEG was not available at the moment of the study. Nevertheless, since we are using a paradigm that we have repeatedly studied in the past and have shown it entrains neural activity and modulates behavior rhythmically, we are confident our results are of interest on their own. For readability of our answers, we numbered to comments below.

1. It would have been interesting to develop the fact that tACS did not "overwrite" neural entrainment to the auditory stimulus. The authors try to explain this effect by mentioning that "tACS is most effective at modulating oscillatory activity at the intended frequency when its power is not too high" or "tACS imposes its own rhythm on spiking activity when tACS strength is stronger than the endogenous oscillations but it decreases rhythmic spiking when tACS strength is weaker than the endogenous oscillations". However, it is relevant to note that the oscillations in their study are by definition "not endogenous" and one can interpret their results as a clear superiority of sensory entrainment over tACS entrainment. This potential superiority should be discussed, documented, and developed.

We thank the reviewer very much for this remark. We completely agree that our results could be interpreted as a clear superiority of sensory entrainment over tACS entrainment. We have now incorporated this possibility in the discussion.

Page 16, line 472-478: “Alternatively, our results could simply be interpreted as a clear superiority of the auditory stimulus for entrainment. In other words, sensory entrainment might just be stronger than tACS entrainment in this case where the stimulus rhythm was strong and salient. It would be interesting to further test whether this superiority of sensory entrainment applies to all sensory modalities or if there is a particular advantage for auditory stimuli when they compete with electrical stimulation. However, answering this question was beyond the scope of our study and needs further investigations with more appropriate paradigms.”

1. The authors propose that "by applying tACS at the right lag relative to auditory rhythms, we can aid how the brain synchronizes to the sounds and in turn modulate behavior." This should be developed as the authors showed that the tACS lags are highly variable across sessions. According to their results, the optimal lag will vary for each tACS session and subtle changes in the montage could affect the effects.

We thank the reviewer for this remark. We believe that the right procedure in this case would be using close-loop protocols where the optimal tACS-lag is estimated online as we discuss in the summary and future directions sub-section. We tried to make this clearer in the same sentence that the reviewer mentioned.

Page 17, line 506-508: “Since optimal tACS phase was variable across participants and sessions, this approach would require closed-loop protocols where the optimal tACS lag is estimated online (see next section).”

1. In a related vein, it would be very useful to show the data presented in Figure 3 (panels b,d,e) for all participants to allow the reader to evaluate the quality of the data (this can be added as a supplementary figure).

Thank you very much for the suggestion. We have added two new supplemental figures (Fig S1 and S2) to show individual data for Fig. 3b and 3e. Note that Fig. 3d already shows the individual data as each circle represents optimal FM-phase for a single participant.

**Reviewer #1 (Recommendations For The Authors):**
Minor comments:"was optimized in SimNIBS to focus the electric field as precisely as possible at the target ROI" It appears that some form of constrained optimization was used. It would be good to clarify which method was used, including a reference.

Indeed, SimNIBS implements a constrained optimization approach based on pre-calculated lead fields. We have added the corresponding reference. All parameters used for the optimization are reported in the methods (see sub-section Electric field simulations and montage optimization). Regarding further specifics, the readers are invited to check the MATLAB code that was used for the optimization which is made available at: https://osf.io/3yutb

"Thus, each montage has its pros and cons, and the choice of montage will depend on which of these dependent measures is prioritized." Well put. It would be interesting to know if authors considered optimizing for intensity on target. That would give the strongest predicted intensity on target, which seems like an important desideratum. Individualizing for something focal, as expected, did not give the strongest intensity. In fact, the method struggled to achieve the desired intensity of 0.1V/m in some subjects. It would be interesting to have a discussion about why this particular optimization method was selected.

The specific optimization method used in this study was somewhat arbitrary, as there is no standard in the field. It was validated in prior studies, where it was also demonstrated that it performs favorably compared to alternative methods (Saturnino et al., 2019; Saturnino et al., 2021). The underlying physics of the head volume conductor generally limits the maximally achievable focality, and requires a tradeoff between focality and the desired intensity in the target. This tradeoff depends on the maximal amount of current that can be injected into the electrodes due to safety limits (4 mA in total in our case). Further constraints of the optimization in our application were the simultaneous targeting of two areas, and achieving field directions in the targets roughly parallel to those of auditory dipoles. Given the combination of these constraints, as the reviewer noticed, we could not even achieve the desired intensity of .1V/m in some subjects. As we wanted to stimulate both auditory cortices equally, our priority was to have the E-fields as similar as possible between hemispheres. Future studies optimizing for only one target would be easier to optimize for target intensity (assuming the same maximal total current injection). Alternatively, relaxing the constraint on direction and optimizing only for field intensity would help to increase the field intensities in the targets, but would lead to differing field directions in the two targets. As an example, see Rev. Fig.1 below. We extensively discuss some of these points in the discussion section: “Are individually optimized tACS montage better?” (Pages 21-22).

Additionally, we added a few sentences in the Results and Methods giving more details about theoptimization approach.

Page 5, lines 115-116: “Using individual finite element method (FEM) head models (see Methods) and the lead field-based constrained optimization approach implemented in SimNIBS (31)”

Page 27, lines 819-822: “The optimization pipeline employed the approach described in (31) and was performed in two steps. First, a lead field matrix was created per individual using the 10-10 EEG virtual cap provided in SimNIBS and performing electric field simulations based on the default tissue conductivities listed below.”

**Author response image 1. sa3fig1:** E-field distributions for one example participant. Brain maps show the results from the same optimization procedure described in the main manuscript but with no constraint for the current direction (top) or constraining the current direction (bottom). Note that the desired intensity of .1 V/m can be achieved when the current direction is not constrained.

The terminology of "high-definition HD" used here is unconventional and may confuse some readers. The paper cited for ring electrodes (18) does not refer to it as HD. A quick search for high-definition HD yields mostly papers using many small electrodes, not ring electrodes. They look more like what was called "individualized". More conventional would be to call the first configuration a "ring-electrode", and the "individualized" configuration might be called "individualized HD".

We thank the reviewer for this remark. We changed the label of the high-definition montage to ring-electrode. Regarding the individualized configuration, we prefer not to use individualized HD as it has the same number of electrodes as the standard montage.

"So far, we have evaluated whether tACS at different phase lags interferes with stimulus-brain synchrony and modulates behavioral signatures of entrainment" The paper does not present any data on stimulus-brain synchrony. There is only an analysis of behavior and stimulus/tACS phase.

We agree with the reviewer. To be more careful with such statement we now modified the sentence to say:

Page 10, lines 303-304: “So far, we have evaluated whether tACS at different phase lags modulates behavioral signatures of entrainment: FM-amplitude and FM-phase.”

"However, the strength of the tACS effect was variable across participants." and across sessions, and the phase also was variable across subjects and sessions.

"tACS-amplitude estimates were averaged across sessions since the session did not significantly affect FM-amplitude (Fig. 5a)." More importantly, the authors show that "tACS-amplitude" was not reproducible across sessions.

Unfortunately, we did not understand what the reviewer is suggesting here, and would have to ask the reviewer in this case to provide us with more information.

References

Kasten FH, Duecker K, Maack MC, Meiser A, Herrmann CS (2019) Integrating electric field modeling and neuroimaging to explain inter-individual variability of tACS effects. Nat Commun 10:5427.Riecke L, Sack AT, Schroeder CE (2015) Endogenous Delta/Theta Sound-Brain Phase Entrainment Accelerates the Buildup of Auditory Streaming. Curr Biol 25:3196-3201.

Riecke L, Formisano E, Sorger B, Baskent D, Gaudrain E (2018) Neural Entrainment to Speech Modulates Speech Intelligibility. Curr Biol 28:161-169 e165.

Saturnino GB, Madsen KH, Thielscher A (2021) Optimizing the electric field strength in multiple targets for multichannel transcranial electric stimulation. J Neural Eng 18.

Saturnino GB, Siebner HR, Thielscher A, Madsen KH (2019) Accessibility of cortical regions to focal TES: Dependence on spatial position, safety, and practical constraints. Neuroimage 203:116183.

Zoefel B, Davis MH, Valente G, Riecke L (2019) How to test for phasic modulation of neural and behavioural responses. Neuroimage 202:116175.